# FTD-tau S320F mutation stabilizes local structure and allosterically promotes amyloid motif-dependent aggregation

Dailu Chen[1,2,5], Sofia Bali [1,2,5], Ruhar Singh[2], Aleksandra Wosztyl [2], Vishruth Mullapudi [2], Jaime Vaquer-Alicea[2], Parvathy Jayan[2], Shamiram Melhem[3], Harro Seelaar [3], John C. van Swieten[3], Marc I. Diamond [2] & Lukasz A. Joachimiak [2,4] ✉

Amyloid deposition of the microtubule-associated protein tau is associated with neurodegenerative diseases. In frontotemporal dementia with abnormal tau (FTD-tau), missense mutations in tau enhance its aggregation propensity. Here we describe the structural mechanism for how an FTD-tau S320F mutation drives spontaneous aggregation, integrating data from in vitro, in silico and cellular experiments. We find that S320F stabilizes a local hydrophobic cluster which allosterically exposes the [306]VQIVYK[311] amyloid motif; identify a suppressor mutation that destabilizes S320F-based hydrophobic clustering reversing the phenotype in vitro and in cells; and computationally engineer spontaneously aggregating tau sequences through optimizing nonpolar clusters surrounding the S320 position. We uncover a mechanism for regulating tau aggregation which balances local nonpolar contacts with long-range interactions that sequester amyloid motifs. Understanding this process may permit control of tau aggregation into structural polymorphs to aid the design of reagents targeting disease-specific tau conformations.

The microtubule-associated protein tau deposits into β-sheet-rich amyloids in over 25 neurodegenerative diseases commonly referred to as tauopathies that include Alzheimer's disease (AD), Corticobasal degeneration (CBD), Picks disease (PiD), and Chronic traumatic encephalopathy (CTE)[1–3]. In sporadic cases of tauopathies, the wild-type (WT) sequence of the microtubule-associated protein tau gene (MAPT) is linked to each disease. In the last 30 years, nearly 50 dementia-associated mutations have been identified in MAPT, linking the genetics of tau to a broader category of dementia associated with tau, Frontotemporal dementia (FTD-tau)[4]. The majority of MAPT mutations fall within the repeat domain region of tau (tauRD)[1], which has been demonstrated to be crucial for microtubule binding and composes the core of tau fibrils as determined by cryo-EM[5–9]. The

mechanism of mutant tau aggregation that causes FTD-tau is not well understood; however, the mutations are thought to directly enhance tau aggregation kinetics, possibly decrease the capacity to bind and stabilize microtubules, and disrupt splicing regulation leading to changes in isoform production[10–15]. There are no available cryo-EM structures of tau fibrils isolated from patients diagnosed with FTD-tau harboring MAPT pathogenic mutations despite these mutations being commonly used in cell and animal models to study tau dysfunction and aggregation. A better understanding of how FTD-tau mutations promote tau dysfunction is required.

The largely disordered nature of monomeric tau makes it refractory to high-resolution structural biology methods. However, evidence from nuclear magnetic resonance (NMR) studies has

[1]Molecular Biophysics Graduate Program, University of Texas Southwestern Medical Center, Dallas, Texas 75390, USA. [2]Center for Alzheimer's and Neurodegenerative Diseases, Peter O'Donnell Jr. Brain Institute, University of Texas Southwestern Medical Center, Dallas, TX 75390, USA. [3]Department of Neurology & Alzheimer Center, Erasmus Medical Center, Rotterdam, Netherlands. [4]Department of Biochemistry, University of Texas Southwestern Medical Center, Dallas, TX 75390, USA. [5]These authors contributed equally: Dailu Chen, Sofia Bali. ✉e-mail: Lukasz.Joachimiak@utsouthwestern.edu

shown that tauRD encodes local structures. The repeat domain contains four imperfect repeats, each separated by a P-G-G-G motif, which readily adopts a β-turn structure. Immediately following the P-G-G-G motif at the end of repeat 2 is the well-characterized amyloid motif, [306]VQIVYK[311] [16–18]. A number of FTD-tau mutations localize to the regions between repeats[19]. The most common *MAPT* mutation employed in models of tau aggregation uses a tau mutant in which the proline at position 301 is substituted to a serine or leucine. This P301S/L mutation is thought to decrease microtubule binding and enhance aggregation kinetics[10,19]. We have previously characterized how mutations at inter-repeat interfaces, such as P301L/S, showing they drive tau aggregation by destabilizing local structures encompassing the [306]VQIVYK[311] amyloid motif and that increases the exposure of the amyloid motif leads to enhanced self-assembly[19]. Several other mutations, including a serine to phenylalanine mutation at position 320 (S320F), are in the middle of a repeat and do not fall at an inter-repeat interface, suggesting they may follow a different mechanism of tau aggregation.

Under normal non-disease conditions, tau is largely disordered[20] and is rather resistant to aggregation. Aggregation studies with recombinant tau are facilitated by inducers such as heparin, RNA, or other polyanions[21,22]. While in cell or animal models, recombinant or patient-derived fibrils (i.e., seeds) are required to induce aggregation. Commonly studied tau disease-associated mutations (i.e., P301S) enhance aggregation propensity but still require the addition of inducers or seeds to aggregate. To date, only the S320F mutation in tau has been shown to aggregate in cells spontaneously[23]. The S320F *MAPT* mutation causes familial frontotemporal dementia with parkinsonism linked to chromosome 17 (FTDP-17)[24]. The proband presented aged 38 with symptoms fitting the clinical diagnoses behavioral variant FTD (bvFTD) and died aged 53[24]. Immunohistochemistry of tau inclusions indicated the presence of "Pick-like" bodies; however, both 3-repeat (3 R) and 4-repeat (4 R) tau were found in the insoluble tau fractions[24], suggesting that the FTD-tau S320F tauopathy may be distinct from that of the 3 R tau isoform-specific Pick's Disease. The S320F mutation is localized far from an amyloid motif, yet it drives spontaneous aggregation of tau. This suggests that S320F may allosterically influence amyloid motif-dependent aggregation, a mechanism distinct from other FTD-tau mutants proximal to amyloids.

Here we integrated in silico, in vitro, and cellular assays to delineate a structural mechanism for the S320F tau mutation that rearranges transient protective interactions to drive spontaneous aggregation. First, we build on prior data and show that S320F drives aggregation in vitro across tauRD fragments and full-length (FL) tau, and also in cells. We perform peptide aggregation experiments using a series of fragments to identify a minimal region that replicates behavior observed in tauRD and FL tau. Using these tau fragments, we employed Molecular Dynamics (MD) simulations to uncover how the S320F destabilizes transient nonpolar contacts and creates new contacts that allosterically expose the amyloid motif. Our data support a model where sequence motifs preceding P-G-G-G in repeats 2 and 3 (R2 and R3, respectively) provide synergistic protection of the amyloid motif [306]VQIVYK[311] and the disengagement of this protection driven by S320F mutation allosterically exposes [306]VQIVYK[311] and enhances self-assembly. Mutations at these new S320F-based interaction sites of contact reverse the aggregation of S320F in vitro in tau peptides, tauRD, and FL tau, as well as in cells. Finally, we employ in silico computational design to stabilize S320F nonpolar interactions further. Our designed sequences aggregate spontaneously in vitro and in cells. Together our work highlights mechanisms to regulate tau assembly and uncovers how an FTD-tau S320F mutation drives tau aggregation. We leverage the knowledge of this mechanism to create new tau sequences that assemble spontaneously. Our data support the concept that tau aggregation can be inhibited or promoted by disrupting or stabilizing transient nonpolar contacts.

## Results

### S320F tau aggregates spontaneously in vitro and in cells

Prior studies on S320F tau have shown that it can form seeds in vitro that can be detected in cell-based models of tau aggregation[23]. To understand the mechanism of S320F-driven tau aggregation, we sought to reproduce this initial observation using 4 R tauRD (herein, tauRD; aa 243–380), 3 R tauRD (aa 243–380 missing 275–305) and FL 2N4R tau under spontaneous aggregation conditions in the absence of polyanionic inducers such as heparin. We produced recombinant FL 2N4R tau, 4 R tauRD, and 3 R tauRD and compared the aggregation propensity of WT, S320F, and a previously characterized FTD-tau P301S mutant[19] (Fig. 1a). We evaluated spontaneous aggregation of our purified proteins in a Thioflavin T (ThT) fluorescence aggregation assay at a series of concentrations. The data were fit to derive $t_{1/2max}$ values, thus facilitating the interpretation of changes in aggregation kinetics. We find that tauRD S320F spontaneously aggregates without any inducer at 62.5 μM (Fig. 1b; $t_{1/2max} = 1.30 \pm 0.07$ h). P301S tauRD also aggregated spontaneously at this concentration, but the ThT signal did not increase until after 72 h. WT tauRD at 62.5 μM did not aggregate within the 96-h duration of the experiment (Fig. 1b). At 25 μM, S320F tauRD again aggregated spontaneously with a $t_{1/2max}$ of $1.49 \pm 0.03$ h but reached a lower maximum ThT signal, while both WT and P301S tauRD did not aggregate. Fibrils were detected by negative stain TEM in the samples of S320F tauRD at 25 μM and 62.5 μM as well as P301S tauRD at 62.5 μM but not in the P301S at 25 μM or any concentrations of WT tauRD (Fig. 1c). Repeating the experiment at 10 μM, S320F tauRD aggregated with slower kinetics relative to the higher concentrations (Supplementary Fig. 1a; $t_{1/2max} = 3.09 \pm 0.32$ h) whereas WT and P301S tauRD did not aggregate (Supplementary Fig. 1a). In addition to experiments with tauRD, we tested FL 2N4R S320F tau and found it to aggregate spontaneously at 10 μM while the FL 2N4R WT tau counterpart did not aggregate (Supplementary Fig. 1b). As both 3 R tau and 4 R tau isoforms were showed to be present in the sarkosyl insoluble fractions of the FTD-tau S320F patient brain[24], we directly tested a concentration titration of 3 R tauRD and 4 R tauRD in a ThT fluorescence assay (Supplementary Fig. 1c). Consistent with prior data[19], we find the 3 R tauRD to be less aggregation prone than 4 R tauRD, with 3 R tauRD spontaneously aggregating only at the highest concentration, 100 μM. The presence or absence of fibrils for tauRD at 10 μM, 2N4R tau, and 3 R tauRD series were confirmed by TEM (Supplementary Fig. 1d).

To test the structural compatibility of spontaneously formed fibrils by in vitro cellular models of tau aggregation, we employed HEK293T tau biosensor cells. The biosensor cells stably express P301S tauRD as fusions to cyan and yellow fluorescent proteins (CFP/YFP) in two separate constructs[25,26]. Aggregation of the fusion constructs, measured as a Förster resonance energy transfer (FRET) signal between the YFP and CFP, occurs when the cells are treated with exogenous tau seeds. Using this system, we observed that treatment with 25 μM S320F tauRD was able to induce tauRD seeding in cells after a 2 h incubation at 37 °C (Supplementary Fig. 1e). Further, the seeding activity was elevated if the cells were treated with the 4 h in vitro incubated S320F tauRD (Supplementary Fig. 1e). WT tauRD did not induce seeding at any incubation time points, consistent with in vitro ThT and TEM quantification of samples (Supplementary Fig. 1e).

Finally, we tested the capacity of S320F tauRD to aggregate spontaneously in cells. WT, S320F, or P301S tauRD were expressed as a C-terminal fusion to mEOS3.2 in HEK293T cells (Fig. 1d). mEOS3.2 is a FRET-compatible photoconvertible fluorescent protein that emits green fluorescence and, after irradiation with UV, will emit red fluorescence. The green and red mEOS3.2 proteins can yield FRET that reports on tau aggregation (Fig. 1d)[27]. Cell lines expressing the WT, S320F, and P301S tauRD-mEOS3.2 constructs were partially photoconverted from green to red suitable for FRET quantification. Cells were fixed and analyzed by flow cytometry to quantify FRET derived

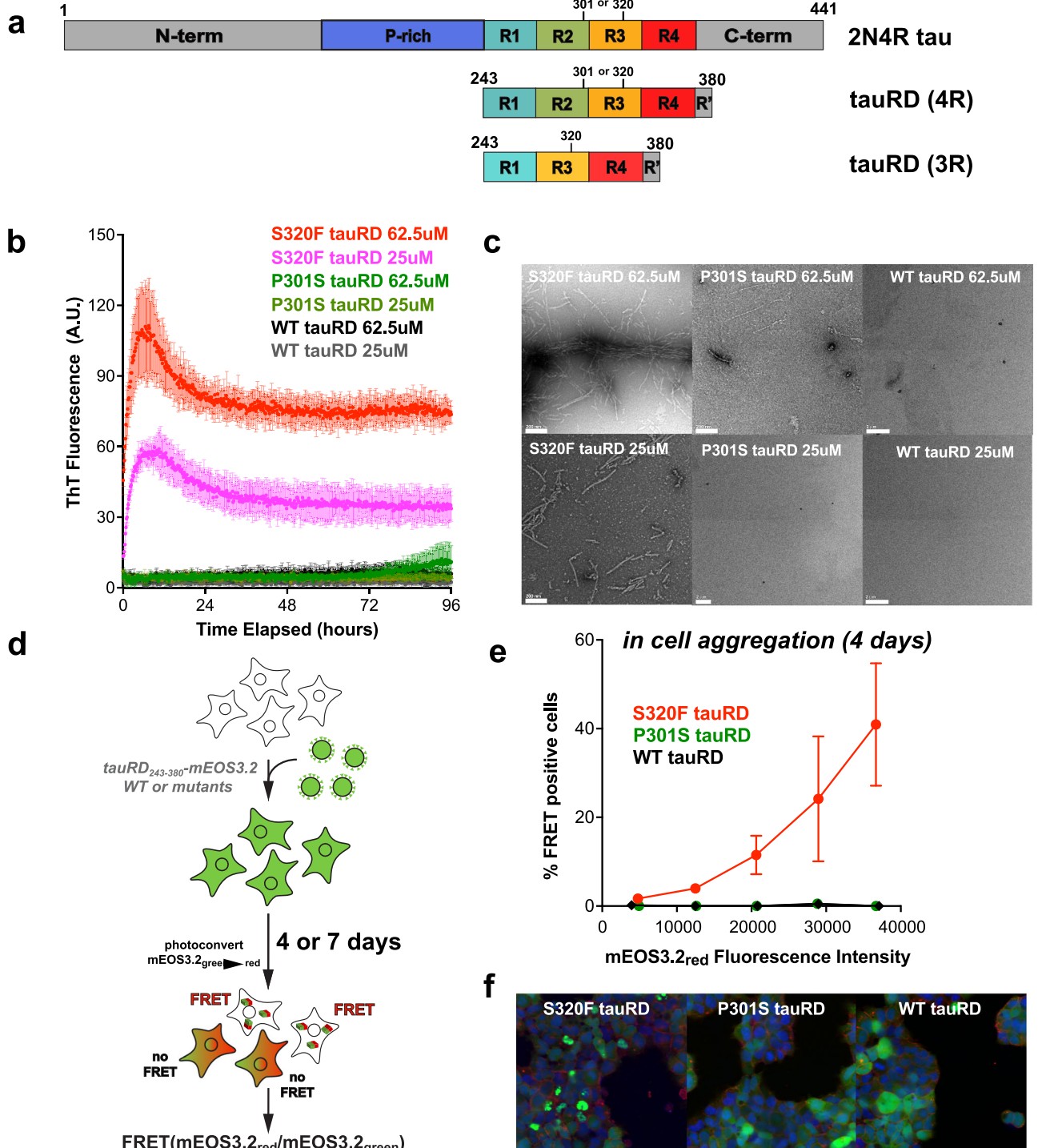

**Fig. 1 | S320F causes spontaneous tau aggregation in vitro and in cells.**
**a** Schematic of 2N4R tau and tauRD where colored as follows: N-term = gray, proline-rich (P-rich) = purple, repeat-1 = cyan; repeat-2 = green; repeat-3 = orange; repeat-4 = red, R' = gray, C-term = gray. Relative positions of 301 and 320 are indicated by black bars. **b** ThT fluorescence assay on S320F (red and magenta), P301S (green and olive), and WT (black and gray) tauRD at 62.5 μM and 25 μM, 37 °C. The data are presented as an average +/− SD for $n = 3$ biological replicates. **c** Representative Transmission Electron Microscopy images of the endpoint ThT assay on S320F, P301S, and WT tauRD at corresponding concentrations. The white bar represents 200 nm distance in 3 images, S320F tauRD 62.5 μM, S320F tauRD 25 μM, and P301S tauRD 62.5 μM, or 2 μm in P301S tauRD 25 μM, WT tauRD 62.5 μM and WT 25 μM. **d** Schematic of cell assay to monitor spontaneous aggregation of WT or mutant tauRD fused to mEOS3.2 in HEK293T cells. HEK293T cells expressing S320F, P301S, or WT tauRD-mEOS3.2 were fixed on Day 4. After photoconversion with UV, a portion of mEos3.2_green was converted to mEOS3.2_red, and FRET could then be detected. **e** FRET (tauRD-CFP/tauRD-mCherry) was measured by flow cytometry on $n = 3$ biological replicates of at least 10,000 cells per condition of S320F (red), P301S (green), or WT (black) after 4 days. The comparison was conducted at multiple fluorescent intensity levels of mEOS3.2_red. Data are shown as averages with 95% CI across three experiments. **f** Representative images of S320F, P301S, and WT tauRD-mEOS3.2 expressed HEK293T cells prior to photoconversion. mEOS3.2 (green), Hoechst33342 (blue, nuclei stain), and Wheat Germ Agglutinin (red, cell membrane stains) fluorescence signals are shown in green and blue, respectively. Scale bar, 15 μm, shown in white.

from the spontaneous aggregation of each tau variant at two time points, days 4 and 7, at each step, ensuring sufficient cells for statistical analysis. We found that S320F tauRD showed an increase in FRET as a function of tauRD-mEOS3.2 expression and photoconversion, indicated by levels of the photoconverted red mEOS3.2 fluorescence signal by gating for different levels of red fluorescence intensity in each experiment (Fig. 1e and Supplementary Fig. 1g). Neither WT nor P301S tauRD-mEOS3.2 constructs showed positive FRET even at high expression and photoconversion levels (Fig. 1e). Similar data was obtained for cells at day 7, although the level of FRET was lower (Supplementary Fig. 1f) possibly due to mild toxicity of the formed aggregates. Representative images of cells expressing S320F, P301S, or WT tauRD-mEOS3.2 showed puncta formation for only S320F while P301S or WT expression remained diffuse in the cell (Fig. 1f). Our data support that the S320F mutation in tau drives spontaneous tau aggregation in vitro and in cells.

### S320F engages with local sequence and controls VQIVYK-based aggregation

In our previous study[19], we identified minimal aggregation regulatory elements in tau based on short peptide fragments encoding inter-repeat fragments (i.e., R2R3, [295]DNIKHVPGGGSVQIVYK[311]) that normally adopt a β-turn stabilized by the P-G-G-G motif. We proposed a model that β-turn stability is inhibitory to aggregation by engaging

with the [306]VQIVYK[311] motif, while destabilization of this structure via mutations (i.e., P301S) promotes assembly[19]. Because S320F is within R3 and downstream of the [306]VQIVYK[311] motif, we hypothesized that the S320F mutation might allosterically disrupt the aggregation-inhibiting β-turn conformation. To identify the key regulatory elements that control S320F aggregation, we designed a series of WT or S320F tau peptide fragments (Fig. 2a). The aggregation behavior of the peptides was evaluated in a ThT fluorescence assay.

We first tested a minimal fragment spanning 316-330, excluding the [306]VQIVYK[311] motif, to determine if the S320F mutation introduces an amyloid motif into the sequence that could directly drive aggregation. Neither $WT_{316-330}$ nor $S320F_{316-330}$ were able to aggregate spontaneously at 200 μM within the time frame of the experiment (Fig. 2b; $t_{1/2max} > 72$ h and Supplementary Fig. 2a) suggesting that the S320F mutation itself did not introduce a novel amyloid motif. In the following fragment spanning 306-324, we shifted the sequence window upstream to include the well-established [306]VQIVYK[311] amyloid motif[16,17]. The $S320F_{306-324}$ and $WT_{306-324}$ fragments aggregated rapidly, preventing accurate capture of the lag or growth phase (Fig. 2b; $t_{1/2max} = 0$ and Supplementary Fig. 2b). The difference in the aggregation behavior of $S320F_{306-324}$ from $S320F_{316-330}$ indicates that the aggregation mechanism of S320F requires the amyloid motif [306]VQIVYK[311]. Aggregation data on these fragments point to an intra-molecular role of S320F in facilitating spontaneous aggregation;

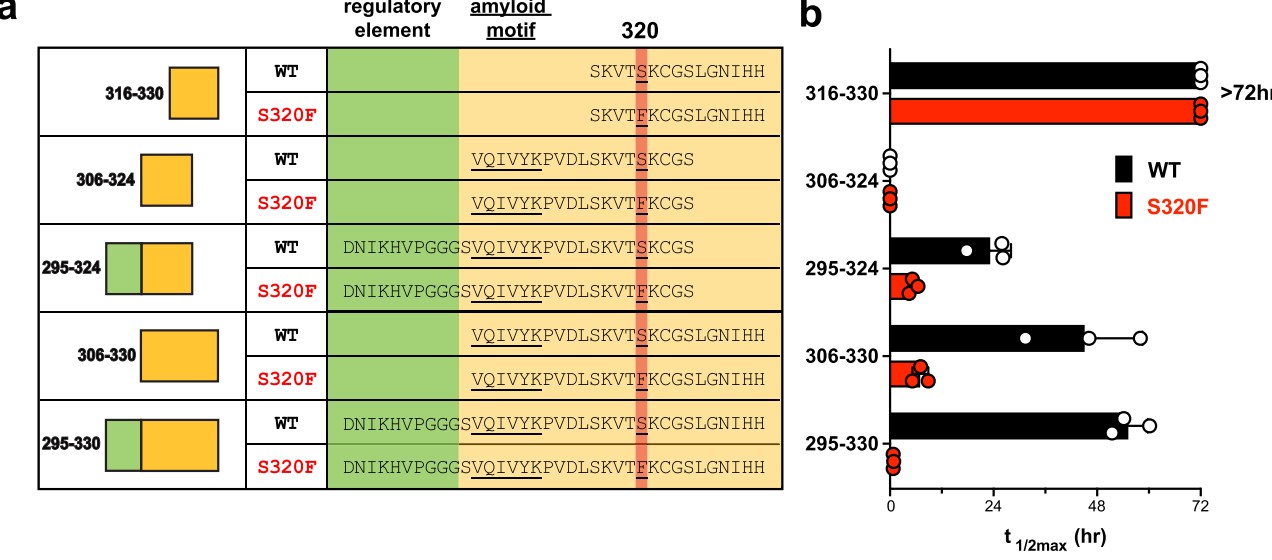

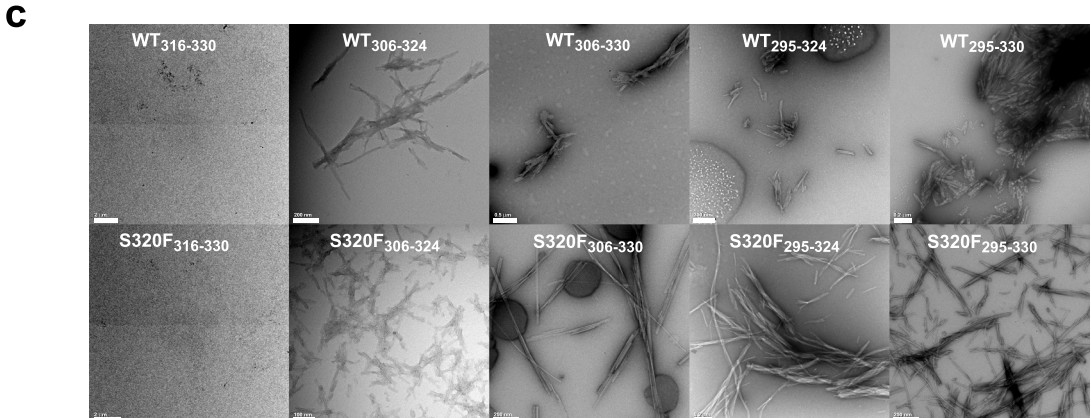

**Fig. 2 | VQIVYK is essential in S320F facilitated aggregation, and the surrounding sequence regulates aggregation. a** Design of sequence fragments encompassing 320 positions in WT or S320F context. The 320 position and [306]VQIVYK[311] are underlined. **b** ThT fluorescence signal at 72 h for each sequence fragment of WT (black) and S320F (red). The data are presented as an average $t_{1/2max}$ +/− SD from fits to a non-linear regression model in GraphPad Prism $n = 3$ biological replicates. **c** Representative TEM images of each peptide from the ThT assays at 72 h endpoint.

additionally, the phenylalanine likely influences [306]VQIVYK[311]-based assembly allosterically.

From our previous study, we concluded that the local sequence upstream of the amyloid motif, [295]DNIKHVPGGGS[305] shields [306]VQIVYK[311] from aggregation[19]. Therefore, we sought to characterize any effects the S320F mutation will have on the upstream sequence. We tested the aggregation behavior of fragments spanning residues 295-324, S320F[295-324], and WT[295-324]. As expected, the aggregation of S320F[295-324] was delayed compared to the shorter S320F[306-324] (Fig. 2b, Supplementary Fig. 2c; $t_{1/2max} = 5.4 \pm 1.0$ h). Surprisingly, WT[295-324] also aggregated, although with slower kinetics than the S320F counterpart (Fig. 2b and Supplementary Fig. 2c; $t_{1/2max} = 23.3 \pm 4.8$ h). This difference raises a new hypothesis that an additional downstream sequence might disturb the protection of [306]VQIVYK[311] from [295]DNIKHVPGGGS[305]. The hydrophobic-rich property of the downstream sequence [312]PVDLSKVTSKCGS[324] might compete for the interaction between [295]DNIKHV[300] and [306]VQIVYK[311]. The addition of another phenylalanine residue in S320F[295-324] might further compete for this interaction, thus explaining the faster aggregation kinetics of S320F[295-324] to its WT[295-324] counterpart.

We then shifted the window of fragment sequence downstream of 320 to 306-330 and found that the aggregation of WT[306-330] was delayed further than WT[295-324] (Fig. 2b and Supplementary Fig. 2d; $t_{1/2max} = 45.2 \pm 13.3$ h). This shift suggests that the downstream sequence [325]LGNIHH[330] might contribute to the amyloid motif protection beyond [295]DNIKHV[300]. As expected, S320F[306-330] was able to aggregate more readily than its WT counterpart (Fig. 2b and Supplementary Fig. 2d; $t_{1/2max} = 7.0 \pm 1.8$ h). However, the more considerable difference in aggregation kinetics between S320F and WT in the context of 306-330 compared to 295-324 implies that the sequence downstream of S320 might play a more substantial role with S320F than the upstream sequence in promoting aggregation.

Given that the protection of [306]VQIVYK[311] with either [295]DNIKHV[300] or [325]LGNIHH[330] can be perturbed by introducing the S320F mutation, we were curious to see how VQIVYK-based aggregation is regulated if both [295]DNIKHV[300] and [325]LGNIHH[330] are present. We designed and tested fragments spanning 295-330. WT[295-330], as expected, had an even further delay in aggregation, suggesting that both ends contribute to the inhibition of VQIVYK-based aggregation (Fig. 2b and Supplementary Fig. 2e; $t_{1/2max} = 55.3 \pm 4.5$ h). On the other hand, with the putative double protection from both ends, S320F[295-330] was able to aggregate with even faster kinetics than S320F[306-330] or S320F[295-324] (Fig. 2b and Supplementary Fig. 2e; $t_{1/2max} = 0.8 \pm 0.1$ h), suggesting that the S320F mutation may negate the [295]DNIKHV[300] and [325]LGNIHH[330]-based protection of [306]VQIVYK[311].

The presence or absence of fibrils at the endpoint of this experiment was shown by negative stain TEM and was consistent with the ThT assays. Across the pairs of different fragments, the largest difference was observed between WT and S320F in fragments 295-330. Moreover, the kinetics of S320F[295-330] seems to recapitulate the behavior of S320F tauRD and S320F 2N4R tau (Fig. 1b and Supplementary Fig. 1b), where no lag phase and an instant elongation phase were observed. Together our peptide fragment experiments imply a structural mechanism where [306]VQIVYK[311] is essential in S320F-based aggregation, and the sequence motifs [295]DNIKHV[300] in R2 and [325]LGNIHH[330] in R3 might synergistically protect the amyloid motif 306VQIVYK311. The introduction of S320F might perturb the protections on [306]VQIVYK[311,] possibly through interaction with the sequence downstream of S320. The increased exposure of [306]VQIVYK[311] may explain the enhanced aggregation of S320F tau, similar to the effects of the P301S mutation.

## S320F allosterically disrupts the shielding of the amyloid motif

To gain molecular insights into the structural mechanism of S320F aggregation behavior, we employed a Molecular Dynamics (MD) approach on WT and S320F tau peptides that comprised the sufficient regulatory elements encoding the entire R3 and the C-terminal portion of R2 spanning residues 295-330 (Fig. 3a, herein WT[295-330] and S320F[295-330]). Simulations were started from the pre-minimized structures and performed as five independent 3 μs trajectories producing ensembles with discernible tertiary conformation features that were not biased by the starting conformation or secondary structure (Supplementary Fig. 3a–c). Comparison of cumulative contact maps derived from merged ensembles (15 μs total per fragment) identified features that distinguish the behavior of WT[295-330] from S320F[295-330] (Fig. 3b, c, and Supplementary Fig. 3d). Focusing on the 320 position in the S320F[295-330], there is a prominent appearance of interactions with [306]VQIVYK[311] that is not present in the WT[295-330]. Additionally, an increased contact to the C-terminal [325]LGNIHH[330] sequence with a break in interactions to residues preceding this sequence (residues 312-315) (Fig. 3b, c, and Supplementary Fig. 3d). This rearrangement of contacts suggests that the S320F mutation might allosterically control VQIVYK-based aggregation. Specifically, S320F might stabilize local structures in the C-terminus, which can disrupt the protective interactions between [325]LGNIHH[330] and [306]VQIVYK[311] and, in turn, disrupt the local N-terminal structure. This implies that [295]DNIKHV[300] and [315]LSKVTS[320] might work synergistically to protect [306]VQIVYK[311], which could explain why S320F[295-330] was able to aggregate even with the inclusion of the previously characterized protector sequence [295]DNIKHV[300] [19]. The structures in the combined 15 μs simulations per fragment were clustered with an RMSD cutoff of 0.6 nm. The mean structure of each of the top five clusters supports our structural analysis of conformations adopted by WT[295-330] and S320F[295-330] (Fig. 3d, e, and Supplementary Fig. 4a, b). Specifically, these more expanded N-terminal local structures were observed in most of the WT[295-330] clusters with the C-terminus folding back and in close contact with the N-terminal local structure (Fig. 3d). These local structures became more modular for S320F[295-330], where C-terminus formed its own local structure separate from the shorter N-terminal local structure (Fig. 3e).

To determine which residues within 325-330 might be important in the interaction with S320F, we plotted the Cα-Cα distance distribution of all residues to the 320 position (Fig. 3f, g for WT[295-330] and S320F[295-330]). WT[295-330] had more evenly distributed distances to S320 for the residues surrounding the 320 position (Fig. 3f; 312-319 and 325-330), while the S320F[295-330] had two residues, L315 and I328, that were distant in sequence to 320, and each presented a distinct population (>15%) that are within 0.2 nm to F320 suggesting the emergence of a specific interaction. We focused on the putative F320-I328 interaction as this position was further from the mutation site, and I328 appeared to interact with [306]VQIVYK[311] in the WT[295-330] ensemble. We hypothesized that hydrophobic contacts between F320 and I328 play an important role in stabilizing alternate interactions that could expose the [306]VQIVYK[311] motif.

To test the role of the F320-I328 interaction, we repeated the MD simulations on the 295-330 fragment encoding S320F and I328S (S320F_I328S[295-330]). Analogous to the WT[295-330] and S320F[295-330] simulations, the replicate simulations for S320F_I328S[295-330] show similar overall features that are not biased by the initial conformation and sample alternate secondary structures (Supplementary Fig. 3e–g). The cumulative contact map and difference maps to WT[295-330] and S320F[295-330] revealed the recovery of WT-like contacts between the C-terminal [325]LGNIHH[330] and both [306]VQIVYK[311] and [295]DNIKHV[300] (Fig. 3h and Supplementary Fig. 3h). The mean structures from the top five clusters observed in the S320F_I328S[295-330] ensemble supported our observation from the cumulative contact map (Fig. 3i). The Cα distance distribution of contacts between I328 and F320 in the S320F_I328S[295-330] fragment were also consistent with the stabilizing hydrophobic interaction between F320 and I328, as the number of structures with residues in close contact (<0.2 nm) with positions 320 and 328 were reduced (Fig. 3j). To quantify this change in interactions

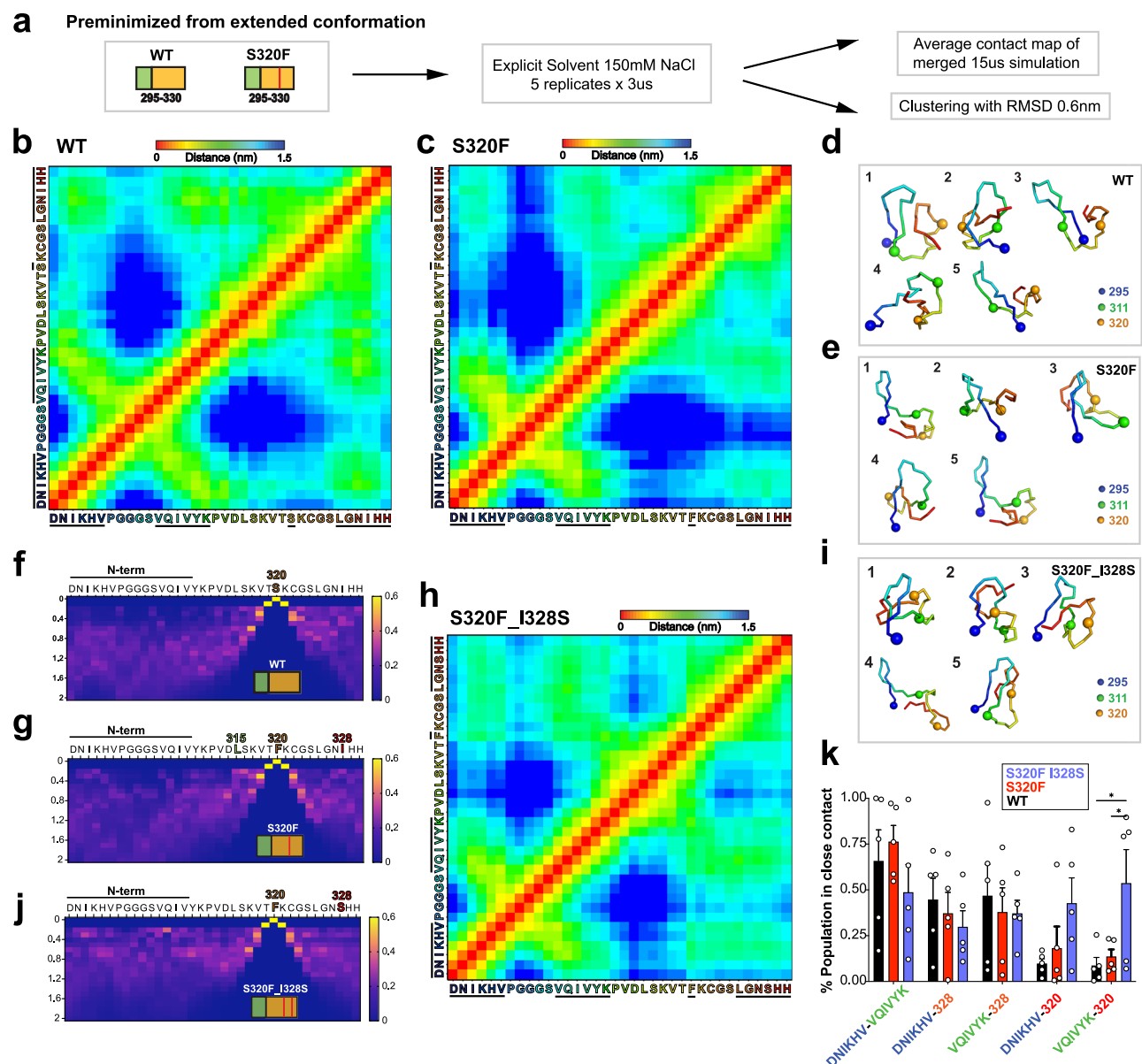

**Fig. 3 | MD simulations reveal different conformations for WT and S320F 295-330 fragments and implicate I328 to be involved in the aggregation mechanism of S320F. a** Simplified schematic of the MD simulation procedure. Average of five replicate simulations from independent trajectories of 3 μs is shown in a contact map for **b** WT$_{295-330}$, and **c** S320F$_{295-330}$. The color bar of the contact map indicates the distance between pairs of residues in the range of 0–1.5 nm. The mean structure of each of the top five clusters within 0.6 nm RMSD for simulations of **d** WT$_{295-330}$, and **e** S320F$_{295-330}$. Positions 295, 311 and 320 are shown in blue-, green- and orange-colored spheres. Sequences are colored with the rainbow spectrum and match the residue coloring in **b** and **c**. Cα distance distribution of each residue to the S320 position in **f** WT$_{295-330}$ and **g** S320F$_{295-330}$ replicate average. The color scheme indicates the % population (scale 0–1) of the residue at a particular distance to 320. Arrows point to the positions/regions of interest. **h** Average of five replicate simulations from independent trajectories of 3 μs for S320F_I328S$_{295-330}$ is shown in a contact map. **i** Top five clusters of 0.6 nm RMSD for simulation on S320F_I328S$_{295-330}$. Atoms are shown as spheres and colored as in **d**. **j** Cα distance distribution of each residue to the S320 position in S320F_I328S$_{295-330}$. **k** % population in close contact (<0.8 nm) for five sets of contacts of interest in WT$_{295-330}$ (black), S320F$_{295-330}$ (red), and S320F_I328S$_{295-330}$ (blue). Data are shown as averages across 5 replicates with standard deviation.

across fragments, we determined the fraction of structures that sampled short distances across replicates (<0.8 nm) between the center of mass of the fragments for five different interactions: DNIKHV to VQIVYK, DNIKHV to 320, VQIVYK to 320, DNIKHV to 328 and VQIVYK to 328 (Fig. 3k). We find that in the majority of the simulations, VQIVYK resides in proximity to DNIKHV and that residue 328 overall decreases proximity to VQIVYK/DNIKHV in S320F and S320F_I328S when compared to WT. By contrast, the simulations of S320F_I328S$_{295-330}$ reveal an increase in interactions between phenylalanine 320 and DNIKHV or VQIVYK (Fig. 3k). In parallel, we directly computed Cα distances between positions 295 to 311, 320 to 328, and 308 to 328 for the mean

structures of each of the top five clusters from the WT$_{295-330}$, S320F$_{295-330}$ and S320F_I328S$_{295-330}$ simulations (Supplementary Fig. 4b, c). The top clusters followed the same trend where the 295 to 311 average distance was relatively short for WT$_{295-330}$, S320F$_{295-330}$ and S320F_I328S$_{295-330}$ (Supplementary Fig. 4d). By contrast, the 320-328 distance was larger in WT$_{295-330}$ and S320F_I328S$_{295-330}$ compared to S320F$_{295-330}$ (Supplementary Fig. 4c). While the C-terminal contacts to 311 were recovered upon introduction of the I328S in the context of S320F (Supplementary Fig. 4c). This can also be explained by the phenylalanine and isoleucine stabilizing interactions in the top clusters for S320F$_{295-330}$ (Supplementary Fig. 4d). We also observe that the 308-

328 (as a proxy for the $^{306}$VQIVYK$^{311}$ to I328) distance was smaller in WT and S320F_I328S$_{295-330}$ (Supplementary Fig. 4d). Finally, we show that similar distance distributions to position 320 can be acquired for WT$_{295-330}$, S320F$_{295-330}$ and S320F_I328S$_{295-330}$ using an alternative force field and that cumulative distance distributions of the 320-328 interaction fall within the standard deviation of the original replicate data (Supplementary Fig. 4e). Consistent with our tau fragment aggregation data (i.e. Fig. 2), our collective MD simulations on WT$_{295-330}$ and S320F$_{295-330}$ identified structural arrangements driven by new nonpolar contacts to S320F. Mutation of a nonpolar residue that contacts S320F reverses this interaction yielding ensembles more similar to WT$_{295-330}$.

### I328S suppresses the effect of S320F-based tau aggregation in a peptide system

Connecting the MD results with the fragment aggregation assay, our data support that I328 interacts with S320F through nonpolar interactions. We hypothesized that mutating I328 to a polar residue such as serine could disrupt this interaction and reduce the aggregation phenotype. We tested this hypothesis directly in vitro in a ThT aggregation assay leveraging the fragment spanning 295-330 as well as tauRD and in cellular models of tau aggregation. We first performed aggregation experiments on S320F_I328S$_{295-330}$, S320F$_{295-330}$, and WT$_{295-330}$ tau fragments. We found that combining S320F with I328S slows down the aggregation kinetics over 10-fold compared to S320F alone while also reducing the maximum ThT fluorescence amplitude (Fig. 4a and Supplementary Fig. 5a, e; $t_{1/2max}$: S320F_I328S$_{295-330}$ = 11.75 ± 0.18 h S320F$_{295-330}$ = 0.79 ± 0.27 h). As a comparison, the S320F_I328S$_{295-330}$ mutant still aggregated faster than the WT control (Fig. 4a; $t_{1/2max}$ = 49.6 ± 2.6 h). From the simulations, we also identified L315 as another potential nonpolar residue that is proximal to S320F that could stabilize these putative local structures. Using our 295-330 peptide system, we tested whether L315S could also suppress the aggregation effect of the S320F mutation. As a control, we tested L325S, which is nonpolar but did not appear to be in close contact with 320F (Fig. 3c). We found that L315S_S320F$_{295-330}$ aggregated more slowly compared to S320F$_{295-330}$ but similarly to S320F_I328S$_{295-330}$. At the same time, the S320F_L325S$_{295-330}$ mutations had little effect on aggregation (Supplementary Fig. 5a, e). These experiments support the idea that nonpolar contacts with S320F observed in the MD simulations play an essential role in stabilizing alternate local structures that allosterically expose $^{306}$VQIVYK$^{311}$ for aggregation.

Following up with the I328S mutation, we hypothesized that the aggregation suppression role of I328S was specific to S320F due to the disruption of local interactions and not simply the removal of a nonpolar side chain. Thus, we tested the effect of I328S on another disease-associated mutant, P301S, which is far from I328, is not predicted to be associated with I328, and, most importantly, directly drives aggregation by exposing $^{306}$VQIVYK$^{311}$ [19]. P301S_I328S$_{295-330}$ was found to aggregate to a comparable ThT signal as P301S$_{295-330}$ despite having moderately slower aggregation kinetics (Supplementary Fig. 5b, e; $t_{1/2max}$: P301S$_{295-330}$ = 6.2 ± 0.3 h and P301S_I328S$_{295-33}$ = 11.6 ± 0.5 h). Therefore, although hydrophobicity at 328 can still play a role in aggregation, the repair effect of I328S is likely specific to the S320F mutant and may be mediated by the disruption of the F320 and I328 interaction. Because hydrophobicity can be important for aggregation, we next tested whether substituting serine at a different position with phenylalanine would produce the same effect and designed a new fragment S324F$_{295-330}$. S324F accelerated the aggregation of 295-330 with $t_{1/2max}$ 33.9 ± 1.0 h compared to WT of $t_{1/2max}$ 49.6 ± 2.6 h, but was much slower than the S320F counterpart of $t_{1/2max}$ 0.79 ± 0.27 h (Supplementary Fig. 5c, e, utilizing the same conditions as in Supplementary Fig. 5a). Although once again, the hydrophobicity at the end of R3 was demonstrated to be important for aggregation, underscoring the crucial role of the 320 position in facilitating aggregation. The

presence or absence of fibrils was confirmed by TEM at the endpoints of ThT experiments and results were consistent with the ThT assay (Supplementary Fig. 5d).

### I328S suppresses the effect of S320F-based aggregation using tau in vitro and in cells

We next wanted to confirm whether the suppression of S320F aggregation effect by the I328S mutation can be translated to tauRD. We produced S320F_I328S tauRD and tested the effect of the I328S mutation to suppress the spontaneous aggregation properties of S320F in a ThT fluorescence aggregation assay compared to tauRD S320F. S320F tauRD again aggregated spontaneously with a $t_{1/2max}$ of 1.9 ± 0.17 h, while S320F_I328S tauRD showed a significantly reduced and delayed ThT signal. At the same time, the WT tauRD control did not aggregate (Fig. 4b). Negative stain TEM of the ThT experiment endpoints confirmed the formation of fibrils in S320F tauRD (Fig. 4c). A small number of fibrils were also detected in the endpoint sample of S320F_I328S tauRD. However, none were found in the WT tauRD sample (Fig. 4c). Additionally, we tested the aggregation propensity of the I328S reversion mutant in an FL 2N4R construct. 40 μM FL 2N4R WT, S320F, and S320F_I328S tau samples were aggregated for 2 weeks; then the endpoint ThT fluorescence signals were compared (Supplementary Fig. 5f). FL 2N4R S320F_I328S tau fluorescence is close to baseline, similar FL 2N4R WT tau and 10-fold lower than FL 2N4R S320F tau (Supplementary Fig. 5f). TEM confirms abundant fibrils for FL 2N4R S320F tau, while only small fragments were seen in FL 2N4R S320F_I328S tau. No fibrils were found in FL 2N4R WT tau (Supplementary Fig. 5g). In parallel, the spontaneous aggregation behavior of S320F tauRD and the repair effect from I328S were tested in the HEK293T cell model of tauRD aggregation leveraging the FRET-compatible photoconvertible mEOS3.2 system. We produced cell lines stably expressing S320F, S320F_I328S, and WT tauRD as fusions to mEOS3.2. The cells were photoconverted, and the cells were analyzed by flow cytometry. To interpret FRET in our cell populations, we employed a gating strategy to compare FRET in cells with similar mEOS3.2 expression levels. As observed previously, S320F tauRD-mEOS3.2 aggregated spontaneously while the S320F_I328S tauRD-mEOS3.2 displayed a >50% drop in FRET-positive cells (Fig. 4d and Supplementary Fig. 5h). The cells expressing WT tauRD-mEOS3.2 showed no FRET. Representative images of cells expressing S320F, S320F_I328S, and WT tauRD-mEOS3.2 showed spontaneous puncta formation for S320F, reduction in the frequency of puncta for S320F_I328S, and diffuse expression of WT tauRD (Fig. 4e). Our data support that the I328S mutation can suppress aggregation activity of the S320F mutant in vitro and in cells. Notably, the peptide fragment experiments can be translated to tauRD and FL 2N4R tau, suggesting that we can capture these local interactions across different length constructs.

To gain insight into the possible conformational changes between WT, S320F, and S320F_I328S tauRD, we employed crosslinking mass-spectrometry (XL-MS). In a previous study[19], we used XL-MS to capture changes in tauRD conformation caused by the P301S mutation and discovered that the disruption of the P-G-G-G alters the stability of the β-turn locally and globally. Here, we extended this method to probe the differences in conformation between WT, S320F, and S320F_I328S using tauRD as a model system. To detect possible changes in conformation, we used disuccinimidyl suberate (DSS), which reacts with primary amines (i.e., lysines). Samples of WT, S320F, and S320F_I328S tauRD were reacted with DSS and quenched, and the monomer species were isolated from an SDS-PAGE gel. The samples were in-gel digested and processed through our XL-MS pipeline[28–30] to identify crosslinked species that include crosslinks (i.e., two peptides linked by DSS) and looplinks (i.e., single peptide dually linked by DSS). In each dataset, the DSS modifications are reported as consensus contacts across five independent samples with a low false discovery rate (FDR) using a high

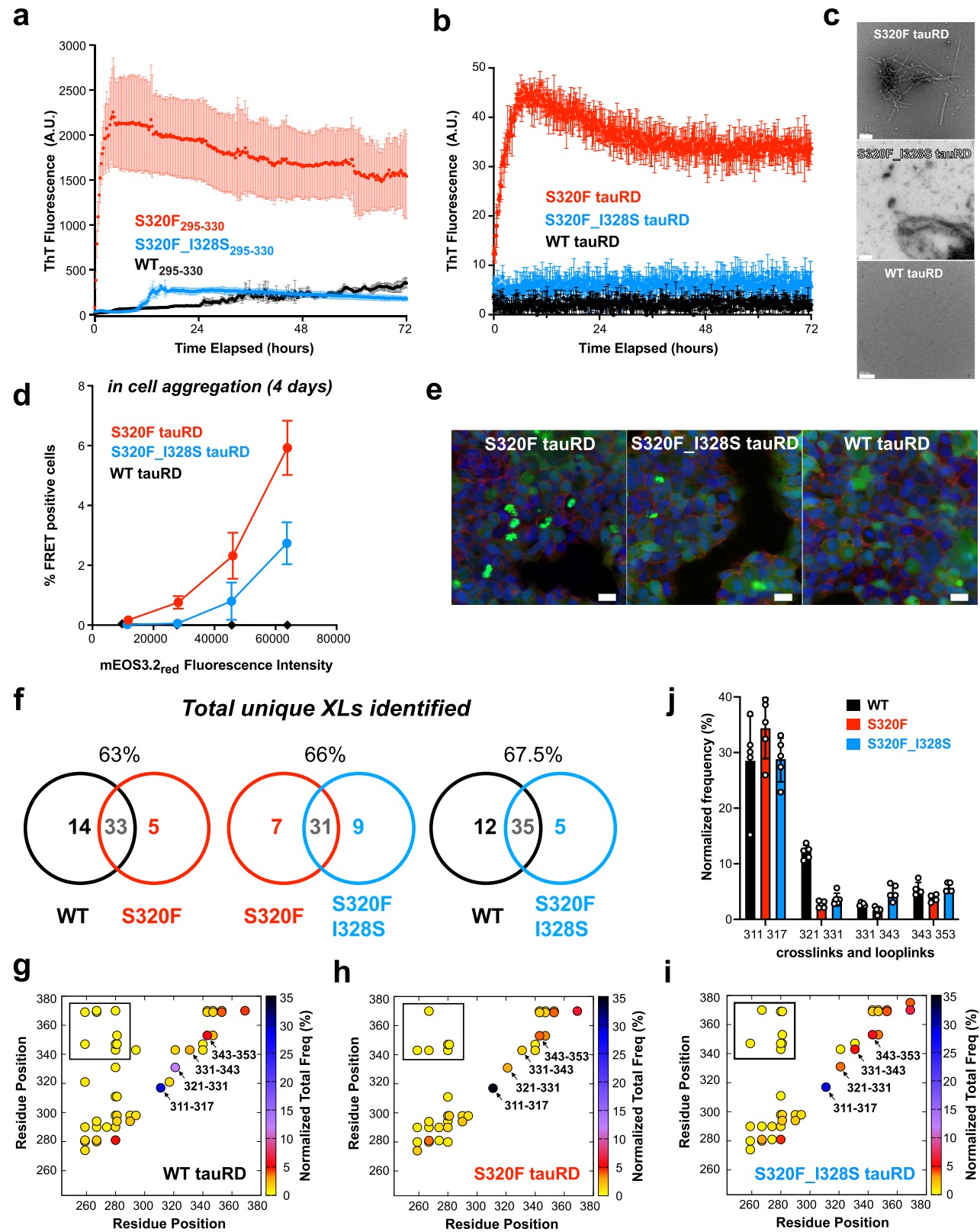

score cutoff (>27). For the WT, S320F, and S320F_I328S tauRD constructs, we observe 47, 38, and 40 consensus crosslinks and looplinks, respectively (Fig. 4f–i). We find 33, 31, and 35 overlapping contacts between the WT-S320F, S320F-S320F_I328S, and WT-S320F_I328S data sets, respectively (Fig. 4f). This translates to 63%, 66%, and 67.5% overlap in contacts, respectively, and indicates that WT has a more similar pattern to S320F_I328S than S320F (Fig. 4f). Comparison of the

crosslink patterns reveals that the S320F_I328S recovers more long-range contacts and is more conformationally similar to WT (Fig. 4g–i, box). Additionally, we find specific contacts within R3 in XL-MS (Fig. 4g–i, arrows) that report on similar contacts observed in MD that varied between $WT_{295-330}$, $S320F_{295-330}$ and $S320F\_I328S_{295-330}$ (Fig. 3k and Supplementary Fig. 4c). Specifically, the contacts between K311-K317 and K321-K331 report as a proxy on the spacing between K311-320

**Fig. 4 | I328S partially repairs aggregation of S320F in vitro and in cells. a** ThT fluorescence assay on S320F$_{295-330}$ (red), S320F_I328S$_{295-330}$ (blue) and WT$_{295-330}$ (black) at 200 μM, 37 °C. **b** ThT fluorescence assay on S320F tauRD (red), S320F_I328S (blue), and WT (black) at 25 μM, 37 °C. Data in (a-b) are presented as an average intensity +/− SD for $n = 3$ biological replicates. **c** Representative TEM images of ThT assay endpoint on S320F, S320F_I328S, and WT tauRD. **d** HEK293T cells expressing S320F, S320F_I328S, or WT tauRD-mEOS3.2 fixed on Day 4. FRET (tauRD-mCerulean/tauRD-mCherry) was measured by flow cytometry on $n = 3$ biological triplicates of at least 10,000 cells per condition on S320F (red), P301S (green), or WT (black). Comparison was conducted at multiple fluorescent intensity levels of mEos3.2$_{red}$. Data are shown as averages with 95% CI across three experiments. **e** Representative images of S320F, S320F_328S, and WT tauRD-

mEOS3.2 expressed HEK293T cells prior to photoconversion. mEOS3.2 (green), Hoechst33342 (blue, nuclei stain), and Wheat Germ Agglutinin (red, cell membrane stains) fluorescence signals are shown in green and blue, respectively. Scale bar,15 μm, shown in white. **f** Number of unique DSS linkages (crosslinks and loop-links) in conditions of tauRD WT, S320F, or S320F_I328S and their overlaps. **g–i** Consensus DSS linkages (circles) are shown in contact maps color-coded by summed frequency across replicates which are normalized by the total number of linkages in each condition. Boxed linkages are defined as long-range contacts. Arrows point to linkage pairs that show differences across the three conditions. **j** Bar plots showing the arrow-pointed pairs from **g–i**. Error bars represent a 95% CI of each condition, $n = 5$ technical replicates. Values from each of the five replicates are shown as white dots.

and 320-328, respectively. We find that K321-K311 is observed 12 ± 1.2 in WT, drops to 2.7 ± 0.6 in S320F and increases towards WT to 3.7 ± 1.1 in S320F_I328S (Fig. 4g–h, j). For the other contacts, we find that the K311-K317 contact in WT is observed 28 ± 8 times while in S320F it increases to 35 ± 5 and in S320F_I328S it drops back down to 29 ± 4 (Fig. 4g–h, j). Other contacts, including K331-K343 and K343-K353 also change but are further away from S320F and thus may have secondary effects to the ones observed closer to position 320. Because tau is intrinsically disordered and the interactions it samples are transient, it is difficult to interpret these changes in XL contact occupancy. However, the crosslinking patterns for WT and S320F_I328S involving local elements are more similar when compared to each other but are comparably distinct from S320F, findings that correlate with the aggregation behavior. Together, our data support that S320F contacts I328 to expose [306]VQIVYK[311] allosterically and drive amyloid assembly and that this interaction can be reversed by introducing the I328S mutation to reduce aggregation in vitro and in cells using tauRD as a model. Moreover, we use XL-MS to probe changes in conformation and identify key contacts that correspond between WT and S320F_I328S and are distinct from those observed in S320F.

## Computational design of nonpolar contacts produces spontaneously aggregating tau sequences

It remains unknown how to control tau folding into discrete fibril conformations. Our data suggest that the S320F mutation allosterically changes how the amyloid motif is exposed, thus enabling conformational changes that lead to spontaneous aggregation. To test this hypothesis more precisely, we wanted to ascertain whether nonpolar substitutions at and in proximity to 320 may lead to a similar aggregation phenotype observed with S320F. FTD-tau driven by the S320F mutation appears to be associated with 3 R and 4 R tau isoforms[24]. Staining FTD-tau S320F tissues with 4 R antibodies shows pathology consistent with perinuclear inclusions Pick-like bodies described in Picks disease or CBD (Fig. 5a). Leveraging the available CBD and PiD tau fibril cryo-EM structures, we wondered whether an S320F mutation could be compatible with these fibril conformations[7,8]. S320 in both the CBD and PiD conformations is placed in a cluster of nonpolar contacts that include L325 and I328 (Fig. 5b and Supplementary Fig. 6a). We utilized these structures to computationally optimize these local nonpolar clusters and test whether they may promote aggregation and stabilize fibrils.

We employed the Rosetta design module to computationally engineer the tau sequence using the CBD and PiD cryo-EM structures as templates (see methods)[31]. For the CBD fibril conformation, we optimized the 320 and 328 positions allowing sampling of all possible amino acid combinations yielding a matrix of 400 designed mutants (Fig. 5b). For the PiD fibril conformation, we combined optimization of residues at 325, 328, and 320 but restricted the search to nonpolar residues to reduce the number of possible solutions (Supplementary Fig. 6a). Our approach allowed us to identify low-energy amino acid substitutions compatible with CBD and PiD fibril backbones (Fig. 5c and Supplementary Fig. 6b). For CBD, two of the top

scoring substitutions involved S320I and S320I with I328V (Fig. 5d, top and middle panels) yielding structures with low energies (Fig. 5d; S320I_I328, ΔREU = −17.7 and S320I_I320V, ΔREU = −19.6) and RMSDs (Fig. 5d; 0.631 Å for S320I_I328 and 0.633 Å for S320I_I320V), compared to the input conformation (Fig. 5d). Similarly, the minimized WT CBD structure retained a low 0.625 Å RMSD relative to the input cryo-EM structure indicating that our method can maintain the correct fibril geometry with near-native side-chain rotamers (Fig. 5d, bottom panel). We performed a similar mutagenesis calculation on the PiD structure and identified S320V as the most optimal solution with low energy and RMSD relative to the WT PiD input structure (Supplementary Fig. 6b; REU = −8.18 and 1.02 Å RMSD).

Based on our hypothesis, we predict that stabilizing these non-polar clusters should yield tau sequences with aggregation propensities similar to S320F. We tested this directly by producing tauRD fragments encoding the designed mutants and measured their spontaneous aggregation properties in vitro. We observed that at 25 μM S320I and S320I_I328V tauRD aggregated readily in an in vitro assay with $t_{1/2max}$ values of 4.4 ± 0.9 h and 5.4 ± 2.0 h, respectively, and similar to S320F tauRD (3.0 ± 0.4 h) while the WT tauRD (>72 h) did not aggregate throughout the experiment (Fig. 5e, f). S320V tauRD at a 25 μM concentration aggregated with a $t_{1/2max}$ of 7.3 ± 1.2 h slightly delayed from the S320F aggregation(Fig. 5f and Supplementary Fig. 6c). Repeating the aggregation experiment using 100 μM reduces the $t_{1/2max}$ for all designed sequences and increases the signal of the ThT fluorescence (Fig. 5f and Supplementary Fig. 6d, $t_{1/2max}$: S320F = 1.2 ± 0.2 h, S320I = 3.0 ± 0.8 h, S320I_I328V = 1.4 ± 0.6 h and S320V = 2.5 ± 0.7 h). Similarly, fast aggregation kinetics were observed for the tau peptides (295-330) with matching mutations (Supplementary Fig. 6e, f, $t_{1/2max}$: WT=>72 h S320F = 29.6 ± 1.6 h, S320I = 7.8 ± 1.2 h, S320I_I328V = 51 ± 3.1 h and S320V = 11.25 ± 3.2 h). For both the CBD and PiD designed tauRD and 295-330, we were able to confirm the presence of fibrils using negative stain TEM (Fig. 5g and Supplementary Fig. 6h). To observe the effect of these mutants on FL 2N4R tau, we compared aggregation of 40 μM FL 2N4R S320I, S320V, S320I_I328V tau and additionally we included FL 2N4R S320I_I328S to test the removal of the hydrophobic partner in the 320-328 interaction. The endpoint ThT fluorescence was compared after 2 weeks of aggregation of all the samples. Similar effects were observed for FL 2N4R as with tauRD. FL 2N4R S320I, S320V, and S320I_I328V tau had ThT fluorescence intensity similar to 2N4R S320F (Supplementary Fig. 6g). By contrast, FL 2N4R S320I_I328S tau mutant had low signal similar to FL 2N4R WT and S320F_I328S tau (Supplementary Fig. 6g). By negative stain TEM, presence fibrils was confirmed for conditions with high ThT fluorescence. In contrast, only a few small fibril fragments were observed in the FL 2N4R S320I_I320S tau sample (Supplementary Fig. 6h). Finally, we tested the behavior of tauRD in our cell-based tau spontaneous aggregation assay. We produced cell lines that express CBD-derived S320I and S320I_I328V tauRD and the PiD-derived S320V tauRD mutants as fusions to mEOS3.2. The cells were grown for 4 days, photoconverted, and analyzed by flow cytometry. Using the same gating strategy as in Fig. 4, we find that the WT and P301S tauRD-

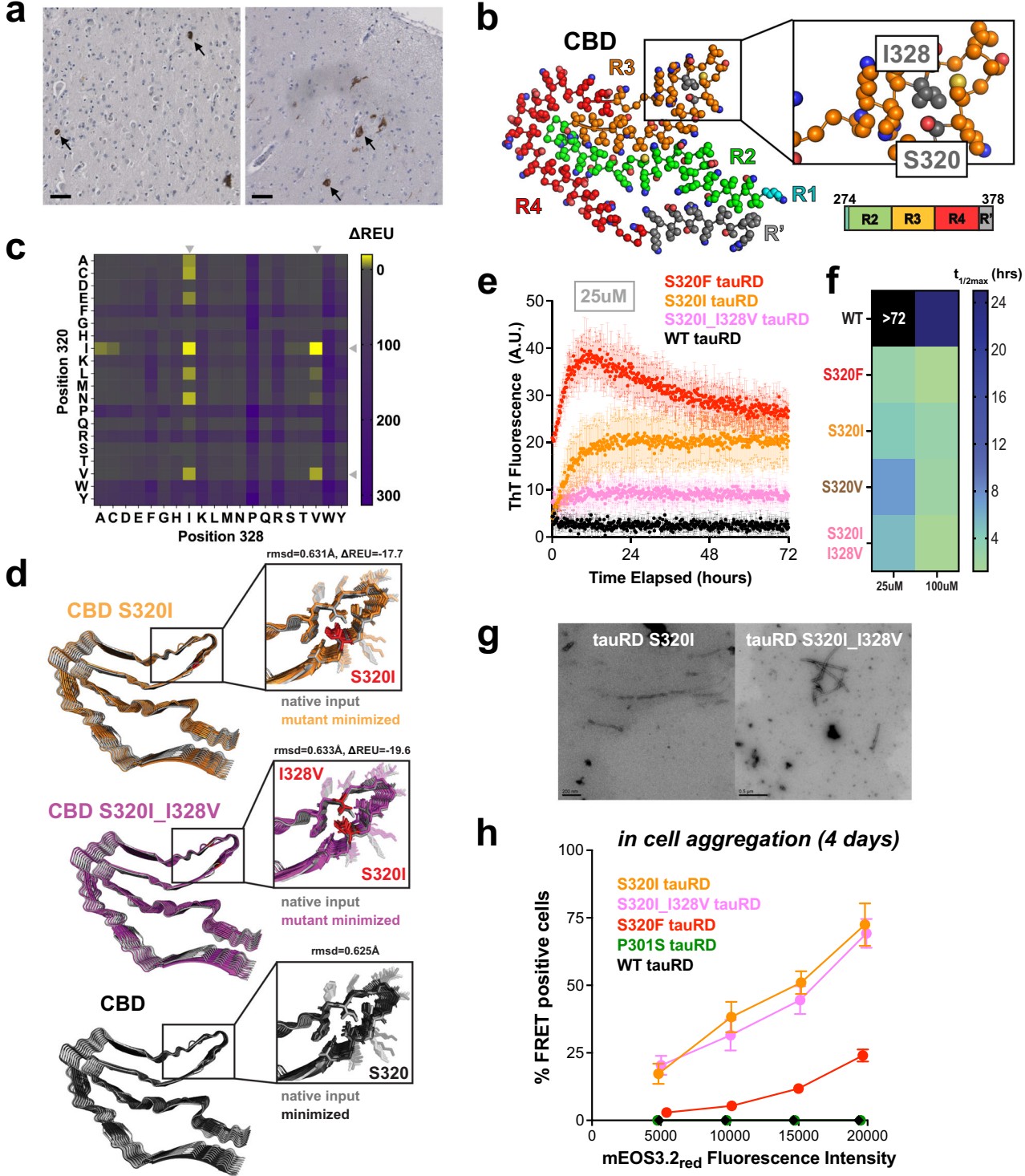

**Fig. 5 | Stabilizing a local hydrophobic cluster promotes aggregation.**
**a** Immunohistochemistry staining of cingulum (left) and hippocampus (right) sections using 4 R antibody. Arrows point to Pick-like bodies. Scale bar, 100 μm, shown in black. **b** Top view, atomic model of the Cryo-EM resolved tau fibril from CBD brains. Residues are colored as in the color scheme of the tauRD schematic. Inset view of the local region encompassing S320 and I328, where S320 and I328 are colored in gray. **c** ΔREU (REU = Rosetta Energy Unit, REU$_{mutant}$−REU$_{WT}$) map for relaxed structures with residues mutated at position 320 and 328 in the context of CBD tau fibril. Color bar indicates the value of ΔREU from negative (yellow) to positive (purple). Gray arrows indicate mutations with favorable energetics.
**d** Representative models of an energy-minimized S320I 9-mer (orange), S320I_I328 9-mer (pink), and WT 9-mer (black) overlaid with the WT native input in the context

of CBD tau fibril structure. **e** ThT fluorescence assay on tauRD S320F (red), S320I (orange), S320I_I328V (pink), and WT (black) at 25 μM, 37 °C. The data are presented as an average +/− SD $n$ = 3 biological replicates. **f** Summary of $t_{1/2max}$ values calculated from fits to a non-linear regression model in GraphPad Prism $n$ = 3 ThT aggregation curves for WT tauRD, S320F, S320I, S320V, and S320I_I328V tauRD experiments performed at 100 uM and 25 uM colored from teal to blue The constructs are labeled as in (**e**). **g** Representative TEM images of the ThT assay endpoints on S320I and S320I_I328V tauRD at 25 μM. **h** FRET (tauRD-mCerulean/tauRD-mCherry) measured by flow cytometry, >10,000 cells per condition, of S320F (red), S320I (orange), S320I_I328V (pink), P301S (green), or WT (black) after 4 days. Comparison was conducted at multiple fluorescent intensity levels of mEOS3.2$_{red}$. Data are shown as averages with 95% CI across three biological replicates.

mEO3.2 constructs remained FRET-negative even at high tauRD expression levels, while cells expressing our designed mutants had a significant FRET-positive population that increased as a function tauRD expression. The CBD- and PiD-derived tauRD mutants (Fig. 5h and Supplementary Fig. 6i) revealed robust in cell spontaneous tau aggregation approaching 75% of cells with FRET positive signal in the population. In comparison, the equivalent expression of S320F tauRD only approached 25% of cells with FRET positive signal. (Fig. 5h and Supplementary Fig. 6i). Our data suggest that the disruption of protective nonpolar contacts drive S320F aggregation properties and that our in silico approach yielded designed tau sequences that mimic S320F by identifying other nonpolar residues that maintain the aggregation-prone phenotype.

## Discussion

In this study, we elucidate the molecular mechanism of how an FTDP-17 S320F *MAPT* mutation drives spontaneous aggregation of tau in vitro and in cells. Leveraging a combination of in silico, in vitro, and cell model experiments, we unveiled the local structural change that led to the spontaneous aggregation behavior of S320F tau. Building upon our previous work[19], we propose that homologous sequence motifs preceding [301]PGGG[304] and [332]PGGG[335] in R2 ([295]DNIKHV[300]) and R3 ([325]LGNIHH[330]) shield the amyloid motif [306]VQIVYK[311] synergistically (Fig. 6). The introduction of the S320F mutation attracts the sequence motif [325]LGNIHH[330] in R3 to form stabilized local hydrophobic cluster that allosterically disrupts the protection of [306]VQIVYK[311], the exposure of which leads to self-assembly of tau (Fig. 6). Understanding the structural basis for S320F-enhanced aggregation enabled us to design sequences beyond the naturally occurring disease-associated mutations that promote rapid assembly. Our data uncover the regulatory nature of transient nonpolar contacts that reduce aggregation in WT tau, which can be disrupted by introducing nonpolar residues that facilitate aggregation.

Our data support that the S320F mutation changes the clustering of nonpolar residues in the monomer towards pro-aggregation contacts and away from anti-aggregation nonpolar contacts which leads to the spontaneous aggregation phenotype. In vitro and in cells, the spontaneous aggregation of S320F is rapid. Perturbation of the pro-

aggregation contacts between S320F and I328 via the mutation of I328S reverses the phenotype in vitro while in cells the percentage of cells with aggregates is reduced. Although the magnitude of the effects is not identical, the trends in vitro and in cells are the same. Indeed, the concentrations and timescales of aggregation differ in vitro and in cells but more importantly the environments of the two reactions are drastically different. We suspect that cellular factors beyond simple crowding effects may play a role. Additionally, our experiments support rearrangements of local structures in the monomeric state that lead to aggregation. We do not have direct structural evidence of the end-stage fibrillar conformation of the S320F mutant; however, we suspect this mutation is compatible with CBD (and PiD) fibril states. Still, future cryo-EM studies on S320F fibrils derived from tau fragments, tauRD and FL tau in vitro and in cells will reveal the details of how the S320F tau monomer misfolding fits into the fibril conformation.

Pro-aggregation nonpolar interaction clusters are seen in the monomer conformation within tau fibril structures determined by cryo-EM, such as residues from [306]VQIVYK[311] (V306, I308, and Y310) form contacts with [373]KLTFRE[378] in AD and CTE or with [337]VEVKSE[342] in PiD and CBD[1,7,8]. Our recent studies demonstrated that these nonpolar clusters are hotspots of interactions important for folding monomers into defined fibril conformations[31]. These interactions might be the driving forces behind tau templating into distinct structural polymorphs. Recently, a study combined MD simulations with experimental data to discover that a fraction of the tauRD ensemble presents local structures[18] matching that of the CBD tau fibril[7]. This sampling suggests that a subpopulation of the monomer can transiently adopt alternative local structures compatible with the fibrillar end product, which may facilitate subsequent intermolecular templating. Importantly, stabilization of these alternate conformations may be sufficient to mediate the conversion of tau into pathogenic fibrils. The position homologous to S320 in R4 is S352, and mutations at this position to leucine or threonine are also linked to FTD-tau[32]. The [337]VEVKSEK[342] sequence in R4 also harbors a motif homologous to [306]VQIVYK[311]. Thus, the modular nature of the four pseudo-repeats suggests that both local and inter-repeat domain interactions may play important regulatory roles. Furthermore, interactions driving the stabilization of these

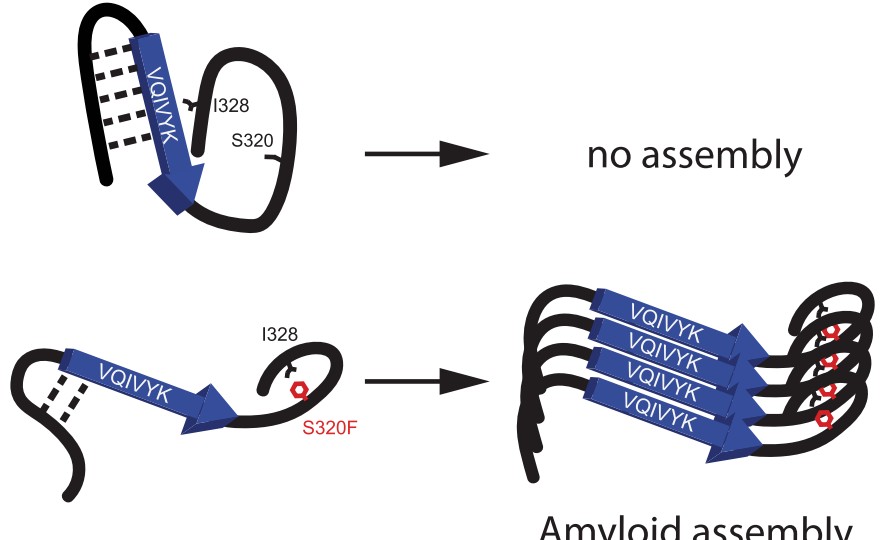

no assembly

Amyloid assembly

**Fig. 6 | Molecular model of the structural differences between WT and S320F within region 295-330 of tau and their subsequent consequences.** WT (top) forms a relatively collapsed structure surrounding the amyloid motif [306]VQIVYK[311], where the sequence motifs at the end of repeat 2 and repeat 3 shield [306]VQIVYK[311] synergistically. In the S320F mutant, the sequence motif at the end of repeat 3 is pulled back to form a local hydrophobic cluster with S320F instead. The loss of the interaction with the sequence motif of repeat 3 also destabilizes the interaction of the sequence motif of repeat 2 with [306]VQIVYK[311]. This may allow a partial exposure or a higher frequency of exposure of the [306]VQIVYK[311] amyloid motif and thus enhances aggregation propensity, which leads to amyloid formation.

alternate conformations in different repeats may hold the key to adopting different tau folds observed in disease. It will be important to validate if these alternate local structures that drive tau aggregation are preserved in the fibrillar state. Future experiments on these sequences must involve structural determination of tau fibrils to link conformational changes in a monomer to the fibril. If this connection is reproduced, it may be possible to control the folding of tau into different conformations by stabilizing these alternate local structures in a monomer through the design of these nonpolar clusters. Furthermore, it may be possible to stabilize these intermediate conformations to aid in the development of diagnostic reagents to detect early stages of pathogenic conformations.

Evidence from other studies demonstrated the importance of how local structures in tau regulate its activity[18]. Indeed, for intrinsically disordered proteins, such as tau, if the correct fragments are used, it is possible to recapitulate the regulatory activity of sequences that limit the exposure of amyloid-forming motifs. Despite a lack of globular structure under normal conditions, these proteins transiently sample alternate conformations, thus limiting the sampling of possible pathogenic states that may predispose exposure of aggregation-prone sequences to promote self-assembly. Combined with more specialized regulatory elements such as β-turn stabilizing motifs, it is possible to limit the aggregation of tau[19] while maintaining biological functions such as microtubule stabilization[33]. A large fraction of pathogenic mutations localize to the repeat domain[1] and thus likely influence the protective local and inter-repeat interactions. A small set of FTD-tau mutations localized outside the repeat domain region and their mechanism of pathogenicity remains largely unknown. For example, it was recently shown that the disease-associated R5L mutation located at the N-terminus of tau induced local structural change, which affected the formation of "tau patches" and hence altered the behavior of tau binding to microtubules, despite being distal from the canonical microtubule-binding region in the repeat domain[33]. Additionally, two FTD-tau mutations, R406W and A152T, lie just outside the repeat domain and their mechanism of pathogenicity remains unknown[34,35]. Thus, even mutations far from the essential core of fibril formation in the repeat domain may alter the distribution of protective transient states that affect tau activity, whether through modulation of aggregation directly or perturbation of other functions such as microtubule stabilization.

This study, together with our previous work[19,36], emphasized the importance of exposure of the amyloid motif [306]VQIVYK[311] in aggregation and strengthened the therapeutic significance of shielding the amyloid motif to mitigate aggregation. Prior work uncovered alternate monomeric states of tau that were aggregation-prone and harbored specific patterns of amyloid motif exposure[36,37] that are detectible early in disease[38]. These insights suggest the possibility of leveraging the exposed amyloid motifs as molecular targets for developing diagnostic and therapeutic interventions. Future development of tauopathy-specific treatments must also consider the diversity of folding states that display distinct patterns of amyloid motif exposure that may begin to accumulate early in disease.

## Methods

### Recombinant full-length tau and tauRD production

We utilized several forms of recombinant tau. The pet28b-tau plasmid encoding full-length 2N4R WT tau with a C-terminal polyhistidine tag was a kind gift from Dr. David Eisenberg (UCLA). The S320F, S320F_I328S, S320I, S320V, S320I_I328V, and S320I_I328S mutants of full-length 2N4R tau in pet28b backbone (also with C-terminal polyhistidine tag) were produced by Twist Bioscience. Each plasmid was transformed into BL21-Gold (DE3) cells. Cells were grown in 1 × Terrific Broth media to OD600 1.4 and induced with 1 mM isopropyl β-D-1-thiogalactopyranoside for 3 h at 37 °C. The cells were harvested and lysed in 50 mM Tris, 500 mM NaCl, 1 mM β-mercaptoethanol, 20 mM imidazole, 1 mM phenylmethylsulfonyl fluoride (PMSF), pH 7.5, using an Omni Sonic Ruptor 400 at 4 °C. The lysates were centrifuged, and the supernatant was applied to a Ni-NTA column and eluted with 50 mM Tris, 250 mM NaCl, 1 mM β-mercaptoethanol, 300 mM imidazole. Eluting fractions containing tau were desalted into 50 mM MES, 50 mM NaCl, 1 mM β-mercaptoethanol (pH 6.0) by PD-10 column (GE). Exchanged fractions were applied to a HiTrap SP HP (GE) and eluted with a 50 mM–1 M NaCl gradient. Fractions containing FL tau were concentrated on an Amicon-15 concentrator, applied to a Superdex 200 Increase 10/300 GL (Cytiva), and eluted into 1× PBS (136.5 mM NaCl, 2.7 mM KCl, 10 mM $Na_2HPO_4$, 1.8 mM $KH_2PO_4$, pH 7.4).

Plasmids for WT tauRD$_{243-380}$ and S320F, S320F_I328S, S320I_I328V, S320V, and S320I mutants in pet28b backbone but without the C-terminal His-tag were produced by Twist Bioscience. WT tauRD and all mutants were expressed the same way as full-length 2N4R tau. Purification procedures for tauRD were adopted from Dr. Paul Seidler (USC). The cells were harvested and lysed in 20 mM 2-(N-morpholino) ethanesulfonic acid (MES), 1 mM (Ethylenediaminetetraacetic acid) EDTA, 1 mM $MgCl_2$, 5 mM β-mercaptoethanol, pH6.8, and appropriate amounts of cOmplete™ EDTA-free Protease Inhibitor Cocktail tablets (Sigma), using an Omni Sonic Ruptor 400 at 4 °C. The lysates were centrifuged, and the supernatant was boiled in a flask with 500 mM NaCl for 20 min in a water bath. The boiled supernatant was centrifuged at $15,000 \times g$ for 15 min. The supernatant, after centrifugation, was dialyzed against a 20-fold volume of 20 mM MES, 50 mM NaCl, 5 mM β-mercaptoethanol, pH 6.8. The dialysis buffer was changed once after 4 h and left overnight. The dialyzed lysate was filtered using a 0.22 μm filter, loaded on a 5 ml HiTrap SP HP (Cytiva), and eluted with a 50 mM–800 mM NaCl gradient in 20 mM MES 50 mM NaCl, 5 mM β-mercaptoethanol, pH6.8. TauRD-containing fractions were concentrated on an Amicon-3 concentrator (EMD Millipore) and applied to a Superdex 75 Increase 10/300 GL (GE) and eluted into 1× PBS (136.5 mM NaCl, 2.7 mM KCl, 10 mM $Na_2HPO_4$, 1.8 mM $KH_2PO_4$, pH 7.4). Aliquots were all stored at −80 °C in 1 × PBS.

### Peptide synthesis

All sequence fragments 316-330, 306-324, 306-330, and 295-330 were synthesized by Genscript with N-terminal acetylation and C-terminal amidation modifications and purified to > 95% purity.

### ThT fluorescence aggregation assays

WT or mutant FL tau and tauRD protein were diluted in 1 × PBS to the desired concentration and filtered with 0.22 μm centrifuge filters before use. A final concentration of 25 μM protein was used for all experiments unless otherwise specified, with 2 mM TCEP and 25 μM ThT. Each sample was vortexed and aliquoted into a 384-well clear bottom plate, each well with 55 uL volume. Peptides were disaggregated as previously described[39]. In brief, lyophilized peptides were dissolved in 200 uL TFA (Pierce) and incubated at room temperature (RT) for 1 h. In a chemical fume hood, the peptide solution was dried under a stream of nitrogen or $CO_2$ gas and then immediately placed under a lyophilizer to remove any residual volatile solvents. The peptide residue was resuspended in 1 × PBS and 2 mM TCEP to a 200 μM concentration to adjust the peptide to buffered reaction conditions, and the sample was adjusted to pH 7 with NaOH. ThT was added to the samples at a final concentration of 25 μM. 55 uL of master mix was added in triplicates in a 384-well clear bottom plate. All conditions were done in triplicates at RT. ThT kinetic scans were run every 10 min on a Tecan Spark plate reader at 446 nm Ex (5 nm bandwidth), 482 nm Em (5 nm bandwidth). The plate was shaken at 800 RPMs for 10 s prior to each data acquisition. Values of the blank wells containing buffer and ThT were subtracted from values of the experimental groups. $T_{1/2max}$ fits for the ThT fluorescence aggregation data were calculated in GraphPad Prism 9.4.1 using the linear regression sigmoidal fit. For 2N4R tau endpoint experiments, the protein was diluted

to 40 uM with 2 mM TCEP and 25 uM ThT. Samples were incubated in Thermomixer at 37 °C with intermittent shaking every 10 min for 10 s at 800 rpm. Readings were taken in FLUOStarOmega At 448 nm Ex (10 nm bandwidth), 482 nm EM (10 nm bandwidth).

## Transmission electron microscopy

An aliquot of 5 μL sample was placed onto glow-discharged Carbon 300-mesh copper grids for 2 min, washed with distilled water for 30 s, and then negatively stained with 2% uranyl acetate for 2 min. Images were acquired on a Tecnai G$^2$ spirit transmission electron microscope (FEI, Hillsboro, OR), serial number: D1067, equipped with a LaB$_6$ source at 120 kV using a Gatan ultrascan CCD camera.

## Seeding assay on tau biosensor cells

Stable HEK293T (ATCC CRL-1268) cell line expressing P301S tauRD-Clover and P301S tauRD-Cerulean (from FM5-CMV)were plated at a density of 35,000 cells per well in a 96-well plate 18–24 h before treatment. 25 μM of WT and S320F tauRD after incubation at 37 °C for various time periods were transduced by Lipofectamine 2000. Specifically, for each well, 10 uL of each condition and 10 uL of transduction reagent [9.5 μL Opti-MEM (Gibco) + 0.5 μL Lipofectamine 2000 (Invitrogen)] were mixed and incubated for 20 min at room temperature. All conditions were done in triplicates with 20 μL total treatment volume applied per well. After 48 h incubation at 37 °C, cells were harvested with 0.05% trypsin and then fixed in 2% paraformaldehyde (Electron Microscopy Services) for 10 min at room temperature, after which PFA was removed, and cells were resuspended in 1 x PBS for flow cytometry analysis.

## Cell expression

FM5-CMV constructs of WT, P301S, S320F, and S320F_I328S tauRD fused to mEOS3.2 at the C-terminal were used for cell expression. Virus of each construct was produced in Lenti-X™ 293 T Cell Line (Takara, Cat. #: 632180). Specifically, 400 ng PSP, 1200 ng VSVG, 400 ng of the plasmid of interest, 7.5 uL TransIT (Mirus Bio), and Opti-MEM (Gibco) were mixed to a final volume of 150 μL and incubated at room temperature for 30 min before adding to the Lenti-X™ 293 T Cell Line. Media (10% FBS, 1% Pen/Strep, 1% GlutaMax in Dulbecco's modified Eagle's medium) containing virus was collected after 48 h and concentrated 50-fold following the protocol of Lenti-X™ Concentrator (Takara, Cat. #: 631232). The concentrated virus in 40 uL volume was added to HEK293T (ATCC CRL-1268) cells plated several hours ahead, starting with 80,000 cells in one well of a 24-well plate. Fluorescence expression was checked 24 h later, and cells were transferred to one well of a 6-well plate upon reaching ~80%–90% confluency. Cells were harvested on Day 7 after virus treatment with 0.05% trypsin and then fixed in 2% paraformaldehyde (Electron Microscopy Services) for 10 min at room temperature, after which PFA was removed, and cells were resuspended in 1 x PBS. Immediately before flow cytometry, cells were photoconverted under UV for 25 min to an optimal ratio of green and red fluorescence for FRET measurements[27].

## Imaging of tau biosensor cell lines

HEK293T cells treated with virus expressing either tau RD S320F, S320F_I328S, or P301S were plated at 10,000 cells per well in media (10% FBS, 1% Pen/Strep, 1% GlutaMax in Dulbecco's modified Eagle's medium) in a 96-well clear bottom plate (Corning, Product # 3603). After 24 h, cells were stained with Hoechst33342 and Wheat Germ Agglutinin at a final concentration of 5 μg/mL in media for 10 min at 37 °C, protected from light. The staining solution was removed and substituted with 1x PBS afterward. The plate was placed on an IN Cell 6000 Analyzer (GE Healthcare) with a heated stage, and 50 fields of view were imaged under 4′,6-diamidino-2-phenylindole (DAPI), and FITC channels at ×60 magnification (Nikon ×60/0.95, Plan Apo, Corr

Collar 0.11–0.23, CFI/60 lambda). Images were exported as TIFF files for downstream analysis.

## Flow cytometry

A BD-LSR Fortessa SORP instrument was used to perform FRET flow cytometry. For the tau seeding assay, mClover, mCerulean, and FRET were measured. To measure mCerulean and FRET signal, cells were excited with the 405 nm laser, and fluorescence was captured with a 405/50 nm and 525/50 nm filter, respectively. To measure mClover, cells were excited with a 488 laser, and fluorescence was captured with a 525/50 nm filter. To quantify FRET, we used a gating strategy where CFP bleed-through into the YFP and FRET channels was compensated using the BD FACSDiva Software. Because some mClover-only cells exhibit emission in the FRET channel, we introduced an additional gate to exclude from analysis cells that exert a false-positive signal in the FRET channel (i.e., false FRET gate). Subsequently, we created a final bivariate plot of FRET vs. Cerulean and introduced a triangular gate to assess the number of FRET-positive cells. This FRET gate was adjusted to biosensor cells that received lipofectamine alone and are thus FRET-negative. This allows for direct visualization of sensitized acceptor emission arising from the excitation of the CFP donor at 405 nm. FRET signal is defined as the percentage of FRET-positive cells in all analyses. For each experiment, 10,000 cells per replicate were analyzed, and each condition was analyzed in triplicates. Data analysis was performed using FlowJo v10 software (Treestar).

For the mEOS3.2 cell expression system, tauRD-mEOS biosensor cells were first photoconverted under UV for 30 min. 10,000 singlet events corresponding to donor (non-photoconverted mEOS3.2) and acceptor (photoconverted mEOS3.2) positive cells were collected for each sample. Collection parameters are detailed in Supplementary Table 1. FCS files were exported from the BD FACSDiva data collection software and analyzed using FlowJo v10 software (Treestar). Compensation was manually applied to correct for donor bleed-through into the FRET channel guided by a sample with non-aggregated and photoconverted tauRD-mEOS. Samples were gated on the acceptor intensity such that cells with similar concentrations of tauRD-mEOS were analyzed to mitigate the contribution of differences in concentration leading to apparent changes in the fraction of FRET-positive cells in each condition. FRET-positive cells were quantified by gating double-positive singlet events with a ratio of FRET to donor signal higher than that of a population of tauRD-mEOS3.2 photoconverted cells without aggregates.

## Gel digested XL-MS sample preparation and mass spectrometry

Preparation of tauRD was diluted in 1 × PBS with 1 mM DTT and cross-linked at a total protein concentration of 60 μM using 300 μg of starting material for WT, S320F, and S320F_I328S tauRD. The cross-linking reaction was initiated by adding DSS stock solution (200 mM DSS-d$_0$ and -d$_{12}$, Creative Molecules, dissolved in DMF) to a final concentration of 10 mM. Samples were incubated at 37 °C for 30 s with 350 RPM shaking. The crosslinking reactions were quenched by addition of ammonium bicarbonate to 100 mM final concentration and incubation at 37 °C 350 RPM for 30 min. Aliquots of the crosslinked samples were resolved on an SDS-PAGE gel, and the monomer band was excised and digested in the gel. In detail, gel pieces were sliced into 1 mm$^2$ cubes and sonicated in 25 mM NH$_4$HCO$_3$/50% ACN (acetonitrile) for 5 min twice, and each time the supernatant was discarded. Gel pieces were then washed with 100% ACN with occasional vortex for 5 min until they shrank and became white. Gel pieces were subsequently evaporated to dryness by lyophilization. Gel pieces were then incubated in 25 mM NH$_4$HCO$_3$ (extra volume to ensure gel pieces were still covered after expanding) with 10 mM DTT at 56 °C for 40 min. After removing the supernatant, gel pieces were incubated in 25 mM NH$_4$HCO$_3$ with 55 mM iodoacetamide and sat in the dark for 30 min at room

temperature. Gel pieces were then washed with 25 mM $NH_4HCO_3$, 25 mM $NH_4HCO_3$ in 50% ACN, and lastly, 100% ACN, each step with intermittent vortexing and sitting for 5 min. Gel pieces were again evaporated to dryness by lyophilization. Total of 60 ug of trypsin (mass spectrometry grade, NEB) was diluted in 25 mM $NH_4HCO_3$ and added to dried gel pieces (3x volume of the gel volume). In gel proteolysis was carried out at 37 °C 600 RPM overnight. Proteolysis solution was saved, and gel pieces were incubated in 5% (v/v) formic acid diluted in $H_2O$ at 37 °C for 10 min, followed by 5 min sonication twice. Samples were then purified by solid-phase extraction using Sep-Pak tC18 cartridges (Waters) according to standard protocols. Samples were evaporated to dryness and reconstituted in water/ACN/formic acid (95:5:0.1, v/v/v) to a final concentration of ~ 0.5 µg/µL. In total, 2 µL each were injected for duplicate LC-MS/MS analyses on an Eksigent 1D-NanoLC-Ultra HPLC system coupled to a Thermo Orbitrap Fusion Tribrid system. Peptides were separated on self-packed New Objective PicoFrit columns (11 cm × 0.075 mm I.D.) containing Magic $C_{18}$ material (Michrom, 3 µm particle size, 200 Å pore size) at a flow rate of 300 nL/min using the following gradient. 0–5 min = 5% B, 5–95 min = 5–35% B, 95–97 min = 35–95% B, and 97–107 min = 95% B, where A = (water/acetonitrile/formic acid, 97:3:0.1) and B = (acetonitrile/water/formic acid, 97:3:0.1). The mass spectrometer was operated in data-dependent mode by selecting the five most abundant precursor ions ($m/z$ 350–1600, charge state 3+ and above) from a preview scan and subjecting them to collision-induced dissociation (normalized collision energy = 35%, 30 ms activation). Fragment ions were detected at low resolution in the linear ion trap. Dynamic exclusion was enabled (repeat count 1, exclusion duration 30 s).

### Analysis of mass spectrometry data
Thermo.raw files were converted to the open.mzXML format using msconvert (proteowizard.sourceforge.net) and analyzed using an in-house version of xQuest[28]. Spectral pairs with a precursor mass difference of 12.075321 Da were extracted and searched against the respective FASTA databases containing tau (TAU_HUMAN P10636-8) or with an S320F or S320F_I328S substitution. xQuest settings were as follows: Maximum number of missed cleavages (excluding the cross-linking site) = 2, peptide length = 5–50 aa, fixed modifications = carbamidomethyl-Cys (mass shift = 57.021460 Da), mass shift of the light crosslinker = 138.068080 Da, mass shift of mono-links = 156.078644 and 155.096428 Da, MS1 tolerance = 10 ppm, MS2 tolerance = 0.2 Da for common ions and 0.3 Da for cross-link ions, search in ion-tag mode. Post-search manual validation and filtering were performed using the following criteria: xQuest score > 27, mass error between −2.2 and +3.8 ppm, %TIC > 10, and a minimum peptide length of six aa. In addition, at least four assigned fragment ions (or at least three contiguous fragments) were required on each of the two peptides in a cross-link. FDRs for the identified cross-links were estimated using xprophet[29]. The five replicate data sets were compared, and only cross-links present in five out of five data sets were used to generate a consensus data set. The nseen (frequency) of each residue position modified by cross-link or loop-link was summed and normalized to the total number of cross-link and loop-link modifications across all residues in each condition. The contact maps were plotted using an in-house Gnuplot script where the color scheme indicates the normalized total frequency in %. The normalized total frequency in % for the four interesting DSS-modified pairs was plotted against the residue positions in Prism with error bars representing a 95% CI.

### Molecular dynamics simulations
Two-stage molecular dynamics simulations were generated, including a preminimization stage and a final simulation stage. Additionally, the final simulation stage was replicated with an alternative forcefield and water model. Preminimization simulations were performed to obtain energetically minimized conformations of an initial extended peptide structure to be used as starting structures for final simulations. Both simulations were performed for three peptides: $WT_{295-330}$, $S320F_{295-330}$, and $S320F\_I328S_{295-330}$. In the case of preminimization simulations, the systems were built based on fully extended conformations of peptides constructed in Pymol. The AMBER99sb-ildn forcefield[40] and TIP3P water model[41] were used for the preminimization stage. The systems were prepared in a dodecahedron box (constructed with a minimum 1.5 nm distance from the edge of the box) and solvated with water (about 30,000 water molecules) and NaCl (150 mM physiological ionic strength). Energy minimization of the constructed setup was performed using the steepest decent algorithm to obtain a maximum force below 1000.0 kJ/mol/nm. Then 10 ns of NVT and 20 ns of NPT (first 10 ns with Berendsen coupling[42] and the last 10 ns with Parrinello-Rahman pressure coupling[43]) equilibrations were performed. The subsequent production level trajectories are based on 2 fs time steps[44]. Production level trajectories were obtained for an NPT ensemble with Parrinello-Rahman barostat and periodic boundary conditions with Particle Mesh Ewald (PME)[45] summation for long-range electrostatics. Five 100 ns simulations per peptide were produced, generating a total of 1.5 µs of simulation for the preminimization stage. These trajectories were analyzed using the GROMACS RMS function—based on the results, extended but energetically minimized conformations were extracted. Final simulations were performed with the same minimization, equilibration, and production workflow with only a few modifications. Here, 10,000 water molecules were placed around the peptide and distant 1.2 nm from the edge of the box. The reduced box size was achieved due to the compact minimized structures and decreasd the runtime for the simulation. A total of 45 µs trajectories were generated through the production of five 3 µs trajectories per peptide. For the replicate simulations, the systems where build with the same conditions but using the CHARMM27 forcefield[46] and TIP4P water model[47]. The same minimization, equilibration, and production workflow was used but only for a single replicate, producing one 3 µs trajectories per peptide for a total of 9 µs of trajectory. All simulations were performed on UTSW's bioHPC computing cluster. All analysis were done using gromacs commands: mdmat, cluster, pairdist, distance, do_dssp, and rms. For distance measurements the region or residue center of mass was calculated, and to calculate percent population in close contact a cuttof of 0.8 nm was used. All plots were generated using inhouse python scripts and GraphPad Prism 9.4.1.

### Immunohistochemistry of S320F tauopathy tissue
Sections of 6 µm were deparaffinized and rehydrated with xylene, a series of ethanol, and water. The antigen retrieval steps were performed in 70% formic acid for 8 min and in 0.01 M sodium citrate for 20 min using an autoclave. Endogenous peroxidase was blocked with 0.3% H2O2 and 0.125% sodium azide. The primary antibody (1:100 Anti-Tau [3-repeat isoform RD3] Antibody, clone 8E6/C11, 05-803, Sigma-Aldrich and 1:100 Anti-Tau [4-repeat isoform RD4] Antibody, clone 1E1/A6, 05-804, Sigma-Aldrich) was incubated overnight at 4 °C in PBS with 0.5% milk (Protifar Nutricia, Zoetermeer, Netherlands) and 0.15% glycine. The sections were washed in PBS and incubated with the secondary antibody (Brightvision Poly-HRP-antiMs/Rb/Ra IgG one component, DPVO-HRP 55, Immunologic) for 1 h at room temperature. Next, incubation with DAB Enhanced liquid system tetrachloride (D3939, Sigma Aldrich) was performed. Finally, the sections were rehydrated, counterstained with hematoxylin, and mounted with Entellan (Rapid Mounting Media for Microscope Slides EMS #14800). All slides were scanned using the Nanozoomer slide scanner (Hamamatsu Nanozoomer 2.0 HT digital slide scanner).

## Computational design of tau sequences using fibril backbones in Rosetta

Using Pymol (version 2.5), fibril PDB structures were created with nine layers using the CBD_T1 fibril (PDB ID: 6tjo) and Pick's Disease (PDB ID: 6gx5) tau fibril structures. Structural alignment was used to superimpose the top two chains with the bottom two chains from a duplicated fibril assembly, preserving the geometry of the assembly while extending the fibril length. Overlapping chains were removed, and chains were renamed to a 9-layer fibril assembly with chain lettering increasing alphabetically from the top to the bottom layer. These assemblies were then used as input for the subsequent mutagenesis and minimization in Rosetta. Changes in assembly energy were calculated using a method adapted from the Flex ddG protocol described by Barlow et al.[48] and more recently expanded for cryo-EM fibril assemblies[31]. From the input assembly, a set of pairwise atom constraints with a maximum distance of 9 Ang are generated with a weight of 1, using the fa_talaris2014 score function. Using this constrained score function, the structure is then minimized. After minimization, the residues within 8 angstroms of the mutation site are subjected to backrub sampling, creating a set of sampled structures capturing backbone variation. These sampled structures are either only repacked and minimized, or the desired mutation(s) is introduced, followed by repacking and minimization. This is repeated for thirty-five independent replicates. The lowest energy bound mutant and bound WT structure from each replicate are extracted, subtracted to give a $\Delta REU^{assembly}_{mut-wt}$, and averaged over the 35 replicates to yield a $\Delta REU^{assembly}_{mut-wt}$ of mutation for the given substitution(s).

$$\Delta REU^{assembly}_{mut-wt} = \frac{1}{n\_replicates} \sum_{replicate} bound_{mut} - bound_{wt} \quad (1)$$

### Statistics and reproducibility

All statistics were calculated using GraphPad Prism 8.0. Three independent ThT experiments were run for each condition. Plots were fitted to a non-linear regression model, from which $t_{1/2max}$ values were derived. $t_{1/2max}$ error represents a 95% CI. TEM grids of endpoint ThT samples were screened and imaged a minimum of 4 times to obtain representative images. Flow cytometry of tau aggregation in cells was conducted in three independent experiments, whose values are plotted as averages with error bars representing a 95% CI.

### Reporting summary

Further information on research design is available in the Nature Portfolio Reporting Summary linked to this article.

## Data availability

All ThT, cell-based aggregation, MD, and XLMS data are available as source data 1, source data 2, source data 3, and source data 4, respectively. Source data is also available on zenodo under accession number 7668320. Raw MS data used for the XL-MS analysis has been deposited in the MassIVE and ProteomeXchange databases under the accession numbers MSV000091047 and PXD040126, respectively. Raw MD trajectories, mass-spectrometry and Rosetta models are available as supplementary data and have been deposited in zenodo under accession number 7668320. PDB ids used in the study include 6tjo and 6gx5. Source data are provided with this paper.

## Code availability

All MD simulations were performed using Gromacs-5.0.4 (available at http://www.gromacs.org). All fibril simulations were performed using Rosetta v3.12 (available at https://www.rosettacommons.org).

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

## Acknowledgements

L.A.J is supported by a Effie Marie Cain Scholarship in Medical Research and by grants from the BrightFocus Foundation (A2019060S), the Chan Zuckerberg Initiative (CZI) Collaborative Science Award (2018-191983), and the Welch Foundation (I-1928-20200401). S.B. was supported by a fellowship from the National Institutes of Health (NIH) NINDS (F31NS127513). Mass spectrometry experiments were carried out at the UTSW Proteomics core. Transmission electron microscopy was performed at the Electron Microscopy Core Facility at UTSW, supported by the NIH (1S10OD021685-01A1 and 1S10OD020103-01). Computational resources were provided by the BioHPC cluster supported by the Lyda Hill Department of Bioinformatics at UTSW. We would like to thank Andrew Lemoff from the UTSW Proteomics Core for his valuable insights on troubleshooting procedures in crosslinking mass spectrometry. We want to thank Vaibhav Bommareddy for preparing the plasmids of CMV FM5 tauRD WT. We want to thank Bryan Ryder for help with TEM. We appreciate Dr. Paul Seidler for sharing the purification protocol on tauRD and the original WT tauRD plasmid. We thank all members of the Joachimiak lab, in particular Valerie Perez and Bryan Ryder, for discussions and input on the manuscript.

## Author contributions

D.C. and L.A.J. initiated the project. D.C. and S.B. purified all proteins involved in the study. D.C., S.B., and P.J. performed all peptide and protein aggregation experiments. D.C. and S.B. performed TEM of tau fibrils. D.C., A.W., and S.B. performed MD calculations, and D.C., A.W., S.B., and R.S. performed all structural analyses. D.C. and J.V.A. carried out cell-based aggregation experiments. D.C. and V.M. performed the Rosetta design calculations on tauopathy fibrils. S.M. performed immunostaining of brain tissues. H.S. and J.C.v.S. provided knowledge on disease genetics and instructions on immunostaining. M.I.D. provided insights into spontaneous cell-based tau aggregation experiments. Finally, D.C., S.B., and L.A.J. conceived of and directed the research as well as wrote the manuscript. All authors contributed to the revisions of the manuscript.

## Competing interests

The authors declare no competing interests.
