## [Peer Review File · Nature Communications]

REVIEWER COMMENTS

Reviewer #1 (Remarks to the Author):

The authors demonstrate that the S320F mutation enables the tau RD, 2N4R tau, and (to a lesser extent) 3R tau RD to aggregate into ThT positive aggregates when allowed to form in PBS supplemented with TCEP. The resulting aggregates from the tau RD with the S320F mutation were also able to seed aggregation when transfected into HEK293T tau biosensor cells stably expressing P301S tau RD fused to CFP/YFP. Additionally, HEK293T cells expressing S320F tauRD fused to mEOS3.2 displayed spontaneous aggregate formation within 4 days while equivalent constructs using P301S or wt tau did not. The authors then establish that the VQIVYK motif is necessary for aggregation, even with the S320F mutation. Elements upstream and downstream of VQIVYK and S320 slow aggregation, but the S320F mutation is able to overcome this protection and still aggregate more quickly than wt peptide in PBS supplemented with TCEP. The authors demonstrate that the increased fibrillization of the 295-330 and tauRD S320F peptides can be selectively mitigated back to wild type-like behavior with a computationally designed I328S or L315S mutation. Addition of the I328S mutant to the S320F mutant in a tauRD-mEOS3.2 construct also displayed reduced aggregation in cells.

Chen and colleagues present an approach to investigate the influence of not only specific mutations in amyloid aggregation, but use computational methods to predict aggregation mitigating mutations and demonstrate that introduction of these mutations has the desired effect in vitro and in cells. This approach goes beyond the commonly asked “what” is tau structure and starts to investigate “why” and “how” is tau structure the way it is?

Overall, the data are clearly presented and the experiments chosen are appropriate to answer the question of how the S320F mutation promotes aggregation. Appropriate controls are included to establish the sequence and structure-specific nature of the observed effects. The conclusions drawn and interpretations made are justified by the data.

Minor comments:

1. In supplementary figure 1g the text in each graph is too small to read.
2. Page 6 paragraph 2: “This suggests that the downstream sequence 25LGNIHH330 might additionally contribute to protection of the amyloid motif beyond 295DNIKHH300.” I believe the 25 before LGNIHH is meant to be 325.
3. In Figure 3 labels for panels d-k are missing.

4. In Figure 3 adding a difference map between b and c would more clearly emphasize the author's conclusions about contact differences between WT and S320F constructs.

5. Page 7 top paragraph / Figure 3d, e, and i: "Specifically, these more expanded N-terminal local structures were observed in most of the WT295-330 clusters with C-terminus folding back and in close contact with the N-terminal local structure (Fig. 3d). These local structures became more modular for S320F295-330, where C-terminus formed its own local structure separate from the shorter N-terminal local structure (Fig. 3e)." I do not see these described differences in Figure 3d and e, both wt and S320F panels show clusters with expanded looking N-termini and a folded back C-terminus. The b and c panels used in Supplementary figure 3 with quantitative measurements are more convincing and should replace Figure 3 d, e, and i in the main text.

6. Discussion paragraph 2: "Recently, a study combined MD simulations

with experimental data to discover that a fraction of the tauRD ensemble presents local structures matching that of the CBD tau fibril7." The cited study (Zhang et al, 2020) does not involve MD simulations and does not support this statement.

7. Page 13 paragraph 2: Tags have influence over expression, purification, and fibrillization of amyloid proteins, so this information would be helpful for researchers that may attempt to reproduce the proteins used in the work shown here. Does the full-length tau construct have a His or other purification tag? If so on which terminus? Was it cleaved prior to aggregation assays?

8. Page 13 paragraph 5: Similar to point 7, was fibrillization carried out under quiescent conditions or shaking?

Reviewer #2 (Remarks to the Author):

In this manuscript Chen et al, combine biophysical and in silico experiments to decipher the local structural changes that lead to the spontaneous aggregation of tau. Using a frontotemporal dementia associated S320F tau mutant, the authors beautifully show that the S320F mutation drives the formation of new hydrophobic contacts. The formation of these contacts disrupts the protecting interaction between R2 and R3, exposing the R3 aggregation-prone region and leading to tau spontaneous assembly.

The study, which contains rigorous characterization of S320F conformations, provides the first clues as to how pathogenic mutations alter tau conformation, ultimately leading to disease. In addition, using this new understanding of the tau aggregation mechanism, the authors were able to go one step further and design new self-assembling tau sequences that aggregated rapidly, not only in vitro, but also in cell lines.

Overall this is a very interesting and timely study that hopefully brings us one step closer to understanding and potentially even controlling pathological tau aggregation.

The main drawback of this manuscript is what would appear to be quite sloppy writing. There are many typos, grammatical errors, and repeating sentences that take away from this experimentally solid study.

Examples just from page 3 -

In the sentence "To date, the S320F mutation in tau has been shown to spontaneously aggregate in cells." the word 'only' is missing

In the sentence:

"The proband presented aged 38 with symptoms fitting the clinical diagnoses behavioral variant FTD (bvFTD), and died aged 53 and died aged 53".

The last 4 words are repeated twice in sequence.

And in the sentence:

"This suggests that the downstream sequence 25LGNIHH330 might additionally contribute to protection of the amyloid motif beyond 295DNIKHV300."

25LGNIHH330 should be 325LGNIHH330

In addition, I have a number of technical questions / comments -

1) Have the authors tested if the fusion of tau to mEOS3.2 protein affects its aggregation rates? This is of special interest, since the fusion is at the C-terminus that is important for the shielding/deshielding process.

2) There are very large differences in the amount and size of fibers generated with the different peptides. Can the authors provide an explanation for these differences? For example why are 306-324 WT fibers significantly longer than 306-330 WT fibers?

3) Full-length tau aggregates much slower than the shorter tau4RD region, suggesting that the N-terminal region also plays a part in the shielding of the aggregation-prone regions. Since the majority of the experiments in the manuscript were done with the shorter tau4RD, I suggest testing the suppression of S320F by the I328S mutation and the newly designed S320I mutant also in the context of the full-length tau proteins. This will fully establish that only the interactions with the C-terminal tau region (and not the N-terminal region) contribute to the regulation of aggregation.

Reviewer #3 (Remarks to the Author):

This study involves investigation of the structural basis for amyloid assembly of tau. Through a systematic series of approaches the authors find evidence for a sequence of tau that masks the accessibility of an amyloid motif through an allosteric mechanism, and for a naturally occurring FTD-mutant, S320F reducing that masking tendency. The work then describes molecular dynamics simulations to build a model for how this allosteric mechanism operates, which is then tested experimentally through a series of new mutations. The findings will likely be of interest to the tau amyloid field.

Overall the study is well-executed and data carefully interpreted. Despite the rest of the study being very solid, I am not convinced that the experiments undertaken in last step of the study unequivocally supports the molecular model – some refinement of the conclusions or perhaps some new experiments might be needed to address this concern, as outlined below (points 3 and 4).

Major points:

1. With respect to the data in Fig 4C, I know it is hard to compare directly, but it is perhaps surprising to see as much aggregation as they did in cells with the I328S mutation because the in vitro data on the peptide shows a much stronger suppression of aggregation. Can the authors comment on this?
2. With the data in Fig 5E, what do the different "extents" of aggregation reflect based on max fluor values for ThT? Have the authors measured solubility by pelleting to better estimate the amount of aggregation?
3. Related to the previous point, the fibrils appear a different morphology to the S320F mutant TEM (Fig. 5f and Supplementary Fig. 5e). Perhaps the fibrils undergo a very different assembly mechanism. Do they cross-seed each other's assembly?
4. The percent of FRET positive cells for the 320V mutant is higher than for the 320F mutant - this enhanced aggregation differs to what was seen with purified protein (extent). How do the authors

explain this? In my mind there are enough discrepancies between the data and model to make me question whether the model is justified.

Other comments:

1. It is not clear how the t50 values were extracted from the data. For example the data in Fig 1b has a complex shape. It is perhaps minor in the grand scheme of things, but it is hard to see how good fits to exponential or sigmoidal curves would be obtained. What is the explanation for the dip in ThT after about 12 hours?
2. The S320F306-324 and WT306-324 fragments form amyloid almost instantly – do these also form fibrillar structures by electron microscopy? It remains possible that the mechanism of fibrillization is different when short peptides are used compared to the intact protein. One way to potentially test this would be to see if the fibrils of the peptides can seed the in-cell aggregation.

Other issues:

1. The meaning of this sentence is unclear: “Most tauopathies are sporadic, linking the wildtype sequence of the microtubule-associated protein tau gene (MAPT) to each disease.” I presume the intended meaning is that wild-type tau forms deposits in the case of sporadic causes of disease.
2. Repeated words in sentence: “and died aged 53 and died aged 53.”
3. Something is missing in the following text as to why this suggests FTD-tau S325F is distinct to Pick’s disease “Immunohistochemistry of tau inclusions indicated presence of “Pick-like” bodies and “diffuse” staining. However, 3-repeat (3R) and 4-repeat (4R) tau were found in the insoluble tau fractions suggesting that the FTD-tau S320F tauopathy is distinct from that of Pick’s Disease.”
4. In this sentence, do the authors mean proximal to amyloid motifs or proximal to amyloids? If the latter, the sentence doesn’t make sense to me. If the former, some editing for clarity is needed. “This may suggest that S320F may allosterically influence amyloid motif-dependent aggregation, a mechanism distinct from other FTD-tau mutants proximal to amyloids.”
5. Bottom of page 8 the authors cite Fig 4a – I think they mean to cite Fig 4b
6. On the top of page 9, the authors cite fibrils in Fig 4b. The electron microscopy (EM) data is not shown in this panel. I presume they mean Fig S4E. I recommend that the EM data be provided in the main figure rather than supp.

Reviewer #4 (Remarks to the Author):

The article “FTD-tau S320F mutation stabilizes local structure and allosterically promotes amyloid motif-dependent aggregation” Chen et al describes an interdisciplinary research to elucidate the role of the pathological S320F mutation in enhancing the aggregation properties of Tau by triggering the exposure of the sequence 306-VQIVYK-311 in an allosteric manner.

The work presents an interesting mechanism that can potentially help understanding the underlying molecular bases of FTD.

Generally the research is conducted at high levels and spans a number of techniques in biophysics to study this pathological mutation.

I have, however, strong concerns on one of the central parts of the study, the MD simulations.

Major points:

1) Studying the properties of IDPs by MD simulations is a challenge, primarily for two reasons: the force field biases and the incompleteness of the samplings. In this study, the authors use AMBER99sb-ILDN. This force field is a decent one for folded proteins but it has some limitations in studying IDP, primarily because it is biased toward secondary structures and compact shapes. In general, MD simulations of IDPs can be heavily biased by the force field propensities (for example see [10.1021/acs.jctc.5b00736](https://doi.org/10.1021/acs.jctc.5b00736)).

In order to make the MD part statistically significant, a second set of simulations of equivalent length should be run using an orthogonal force field. For example CHARMM22* with TIP4P-D waters is a good setup to reproduce NMR data of IDPs. A comparison between AMBER99sb and CHARMM22* simulations must therefore lead to the same conclusions in order to provide evidence for absence of force-field biases in the MD part.

2) The presented simulations aren't assessed for their convergence. If the simulations are not convergent, in addition to the force field dependencies, the results can be biased by the starting configurations as well as by the stochastically sampled non-converged conformations. Specific analyses are therefore required to check the convergence of the phase space exploration in the simulations. Moreover, the contact maps should also be calculated for the individual 5 simulations in each system (i.e. and not only as cumulative). This should prove convergence across the independent simulations.

3) The contact maps show some antiparallel pairings between consecutive stretches in the simulated Tau fragment (e.g. between DNIKHV and GGSVQI). These can be due to populations of beta-hairpin conformations in the ensemble, a known bias of force fields like AMBER99sb-ILDN. A plot of the secondary structure along the trajectories can elucidate this aspect (e.g. just by using `do_dssp` in GROMACS). If this plot shows some conformations featuring beta-hairpins, further analyses are required to validate the accuracy of the simulations. For example by comparing the results with experimental data in literature (e.g. secondary shifts in solution NMR of this fragment, if available).

4) Another important test is to check for possible violations of the periodic boundary conditions. I understand that it is necessary to reduce as much as possible the box size for producing long runs, however, compact structures obtained after equilibration might easily expand during the simulations and therefore violate PBC.

Minor points

5) How were the termini of the peptides treated in the simulations?

6) Labels on figure 3 are partially missing.

We are very grateful to the reviewers for their critical and detailed evaluation of our study. We also thank the reviewers for their positive assessment of our work and helpful suggestions. Please find below our point-by-point response addressing reviewer comments in full to improve and streamline the final manuscript. We hope the reviewers will find this new manuscript version suitable for publication in Nat Comm.

REVIEWER COMMENTS

Reviewer #1 (Remarks to the Author):

The authors demonstrate that the S320F mutation enables the tau RD, 2N4R tau, and (to a lesser extent) 3R tau RD to aggregate into ThT-positive aggregates when allowed to form in PBS supplemented with TCEP. The resulting aggregates from the tau RD with the S320F mutation were also able to seed aggregation when transfected into HEK293T tau biosensor cells stably expressing P301S tau RD fused to CFP/YFP. Additionally, HEK293T cells expressing S320F tauRD fused to mEOS3.2 displayed spontaneous aggregate formation within 4 days, while equivalent constructs using P301S or wt tau did not. The authors then establish that the VQIVYK motif is necessary for aggregation, even with the S320F mutation. Elements upstream and downstream of VQIVYK and S320 slow aggregation, but the S320F mutation is able to overcome this protection and still aggregate more quickly than wt peptide in PBS supplemented with TCEP. The authors demonstrate that the increased fibrillization of the 295-330 and tauRD S320F peptides can be selectively mitigated back to wild type-like behavior with a computationally designed I328S or L315S mutation. Addition of the I328S mutant to the S320F mutant in a tauRD-mEOS3.2 construct also displayed reduced aggregation in cells.

Chen and colleagues present an approach to investigate the influence of not only specific mutations in amyloid aggregation, but use computational methods to predict aggregation mitigating mutations and demonstrate that introduction of these mutations has the desired effect in vitro and in cells. This approach goes beyond the commonly asked “what” is tau structure and starts to investigate “why” and “how” is tau structure the way it is?

Overall, the data are clearly presented and the experiments chosen are appropriate to answer the question of how the S320F mutation promotes aggregation. Appropriate controls are included to establish the sequence and structure-specific nature of the observed effects. The conclusions drawn and interpretations made are justified by the data.

We thank the reviewer for their positive assessment of our study.

Minor comments:

1. In supplementary figure 1g the text in each graph is too small to read.

We thank the reviewer for pointing this out. We have rescaled panel g in supplementary Figure 1 to be larger to improve readability.

2. Page 6 paragraph 2: “This suggests that the downstream sequence 25LGNIHH330 might additionally contribute to protection of the amyloid motif beyond 295DNIKHH300.” I believe the 25 before LGNIHH is meant to be 325.

We thank the reviewer for pointing this out. We have corrected the numbering of this fragment in the text.

3. In Figure 3 labels for panels d-k are missing.

We thank the reviewer for pointing this out. We have included missing panel labels for panels d-k in Figure 3.

4. In Figure 3 adding a difference map between b and c would more clearly emphasize the author’s conclusions about contact differences between WT and S320F constructs.

We thank the reviewer for highlighting this. We have calculated the difference contact maps to help illustrate differences between the ensembles for WT₂₉₅₋₃₃₀, S320F₂₉₅₋₃₃₀ and S320F_I328S₂₉₅₋₃₃₀. We have included a description of the regions that differ between the plots in the results on page 7 and Supplementary Fig. 3d,h.

5. Page 7 top paragraph / Figure 3d, e, and i: “Specifically, these more expanded N-terminal local structures were observed in most of the WT₂₉₅₋₃₃₀ clusters with C-terminus folding back and in close contact with the N-terminal local structure (Fig. 3d). These local structures became more modular for S320F₂₉₅₋₃₃₀, where C-terminus formed its own local structure separate from the shorter N-terminal local structure (Fig. 3e).” I do not see these described differences in Figure 3d and e, both wt and S320F panels show clusters with expanded looking N-termini and a folded back C-terminus. The b and c panels used in Supplementary figure 3 with quantitative measurements are more convincing and should replace Figure 3 d, e, and i in the main text.

We thank the reviewer for this suggestion. We have moved panels from Supplementary Figure 3b to Figure 3d,e, i to more easily illustrate how the contacts are preserved in the structural clusters.

6. Discussion paragraph 2: “Recently, a study combined MD simulations with experimental data to discover that a fraction of the tauRD ensemble presents local structures matching that of the CBD tau fibril⁷.” The cited study (Zhang et al, 2020) does not involve MD simulations and does not support this statement.

We thank the reviewer for pointing this out. We have corrected the reference to Stetzi et al. (PMID 35373198)

7. Page 13 paragraph 2: Tags have influence over expression, purification, and fibrillization of amyloid proteins, so this information would be helpful for researchers that may attempt

to reproduce the proteins used in the work shown here. Does the full-length tau construct have a His or other purification tag? If so on which terminus? Was it cleaved prior to aggregation assays?

We thank the reviewer for pointing this out. The full-length 2N4R tau construct (PET29b) has a polyhistidine tag on the C-terminus. Experiments with tauRD have no tags, and all peptide experiments are carried out with N-terminal acetylation and C-terminal amidation. We have clarified these details in the methods.

8. Page 13 paragraph 5: Similar to point 7, was fibrillization carried out under quiescent conditions or shaking?

We thank the reviewer for pointing this out. Shaking at 800 RPM for 10-sec was carried out prior to each time point acquisition but was not continuous. We have clarified these details in the methods on page 14.

Reviewer #2 (Remarks to the Author):

In this manuscript Chen et al, combine biophysical and in silico experiments to decipher the local structural changes that lead to the spontaneous aggregation of tau. Using a frontotemporal dementia associated S320F tau mutant, the authors beautifully show that the S320F mutation drives the formation of new hydrophobic contacts. The formation of these contacts disrupts the protecting interaction between R2 and R3, exposing the R3 aggregation-prone region and leading to tau spontaneous assembly.

The study, which contains rigorous characterization of S320F conformations, provides the first clues as to how pathogenic mutations alter tau conformation, ultimately leading to disease. In addition, using this new understanding of the tau aggregation mechanism, the authors were able to go one step further and design new self-assembling tau sequences that aggregated rapidly, not only in vitro, but also in cell lines.

Overall this is a very interesting and timely study that hopefully brings us one step closer to understanding and potentially even controlling pathological tau aggregation.

The main drawback of this manuscript is what would appear to be quite sloppy writing. There are many typos, grammatical errors, and repeating sentences that take away from this experimentally solid study.

We thank the reviewer for pointing out grammatical errors and typos. We carefully reviewed the manuscript and fixed all the typos/errors.

Examples just from page 3 -

In the sentence “To date, the S320F mutation in tau has been shown to spontaneously aggregate in cells.” the word ‘only’ is missing

We thank the reviewer for pointing this out. We have corrected this sentence.

In the sentence:

“The proband presented aged 38 with symptoms fitting the clinical diagnoses behavioral variant FTD (bvFTD), and died aged 53 and died aged 53”.

The last 4 words are repeated twice in sequence.

We thank the reviewer for pointing this out. We have corrected this sentence.

And in the sentence:

“This suggests that the downstream sequence 25LGNIHH330 might additionally contribute to protection of the amyloid motif beyond 295DNIKHV300.”

25LGNIHH330 should be 325LGNIHH330

We thank the reviewer for pointing this out. We have corrected this sentence.

In addition, I have a number of technical questions / comments -

1) Have the authors tested if the fusion of tau to mEOS3.2 protein affects its aggregation rates? This is of special interest, since the fusion is at the C-terminus that is important for the shielding/deshielding process.

We thank the reviewer for raising this question. We recognize the limitation of using fluorescent tags to study tau aggregation. In our experience, C-terminal fusions of a number of fluorescent proteins to tauRD do not interfere with tau amyloid assembly in cellular systems of tau aggregation (PMID 31175300, 29988016, 35750209, 34504072). Prior studies have also indicated that fluorescent tags on tauRD do not interfere with its prion-like propagation of tau (PMID 24857020, 27974162). Even so, unequivocal structural data showing that the input conformation is identical to the propagated conformation are currently not available but forthcoming. However, in recent work from our lab, we have developed a computational method to predict energetically critical hotspots specific to each tau structural polymorph. We identified networks of hotspot residues that are important for the stability of different fibrils. We then collaborated with the lab of Dr. Marc Diamond to validate the predicted hotspot sites in cellular assays of tau aggregation. Of the 10 predicted sites in the corticobasal degeneration (CBD) fibril conformation, 8 were validated to be essential for seeding with a CBD strain (see figure below). Suggesting that the tauRD construct can not only form fibrils in cells but can propagate a defined fibril conformation (preprint available on biorxiv <https://doi.org/10.1101/2022.07.01.498342>). Additionally, derived from this study, the S320F mutation promotes spontaneous aggregation of tau in cells. We are in the process of determining structures of recombinant peptides, tauRD, FL tau, cell-derived fibrils, and patient-derived tissues to compare fibril conformations from different fragments and sources directly.

[REDACTED]

2) There are very large differences in the amount and size of fibers generated with the different peptides. Can the authors provide an explanation for these differences? For example why are 306-324 WT fibers significantly longer than 306-330 WT fibers?

We thank the reviewer for pointing this out. We are careful not to over-interpret the TEM images of in vitro reactions beyond demonstrating that fibrils can be found in the samples. In this specific instance, the WT 306-324 and WT 306-330 peptides yield different ThT amplitudes,

with the longer peptide giving a lower fluorescence signal than the shorter peptide – this likely reflects the affinity of ThT for fibril surfaces. Regarding the TEM images, we apologize because, upon closer inspection, the WT306-330 image in Fig. 2 was collected at a slightly different magnification (0.5um vs. 0.2um for others). Overall, we see that the WT fragments yield shorter and less defined fibrils than the S320F.

3) Full-length tau aggregates much slower than the shorter tau4RD region, suggesting that the N-terminal region also plays a part in the shielding of the aggregation-prone regions. Since the majority of the experiments in the manuscript were done with the shorter tau4RD, I suggest testing the suppression of S320F by the I328S mutation and the newly designed S320I mutant also in the context of the full-length tau proteins. This will fully establish that only the interactions with the C-terminal tau region (and not the N-terminal region) contribute to the regulation of aggregation.

We thank the reviewer for this suggestion. Assessing the effects of spontaneous aggregation in the contexts of the FL 2N4R tau protein would be a great test to determine how the interactions identified in the molecular dynamics translate to the most physiologically relevant tau construct. We expressed and purified FL 2N4R WT, S320F, S320F I328S, and S320I tau constructs to test this directly. We also included the hydrophobic mutant S320V and hydrophobic matched double mutant S320I_I328V and a new mutant S320I_I328S. The samples were aggregated for 2 weeks, and ThT fluorescence endpoints were measured (Supplementary Figure 5f and 6g). In agreement with our experiments with the tauRD (and peptides), we observed that the double mutant S320F_I328S had a reduced signal that was more comparable to FL 2N4R WT tau. The results also show that the FL 2N4R mutants S320I and 2N4R S320V aggregated to the same final intensity as S320F. We also included double mutants S320I_I328V and S320I_I328S to test the hydrophobic interaction between the 320 and 328 positions; excitingly, we saw that the matched hydrophobic construct S320I_I328V had even higher ThT intensity as S320F and the mutant S320I_I328S inhibited aggregation similar to S320F_I328S. Finally, we confirmed the presence of fibrils by TEM in the ThT-positive samples and the lack of fibrils in the ThT-negative samples (Supplementary Figure 5g and 6h). Our results on FL tau suggest that the same interactions in the short fragments (295-330) and tauRD are preserved and similarly regulated by changing local hydrophobicity. We have included the description of this data in the results on pages 9 and 11.

Reviewer #3 (Remarks to the Author):

This study involves investigation of the structural basis for amyloid assembly of tau. Through a systematic series of approaches the authors find evidence for a sequence of tau that masks the accessibility of an amyloid motif through an allosteric mechanism, and for a naturally occurring FTD-mutant, S320F reducing that masking tendency. The work then describes molecular dynamics simulations to build a model for how this allosteric mechanism operates, which is then tested experimentally through a series of new mutations. The findings will likely be of interest to the tau amyloid field.

Overall the study is well-executed and data carefully interpreted. Despite the rest of the study being very solid, I am not convinced that the experiments undertaken in last step of

the study unequivocally supports the molecular model – some refinement of the conclusions or perhaps some new experiments might be needed to address this concern, as outlined below (points 3 and 4).

Major points:

1. With respect to the data in Fig 4C, I know it is hard to compare directly, but it is perhaps surprising to see as much aggregation as they did in cells with the I328S mutation because the in vitro data on the peptide shows a much stronger suppression of aggregation. Can the authors comment on this?

We thank the reviewer for pointing this out. The kinetics of tauRD S320F aggregation is very fast in vitro (hours), but in cells, the accumulation of FRET-positive puncta is relatively slow (days). The introduction of I328S in the context of S320F reduces the ThT signal in vitro and reduces the number of FRET-positive cells at 4 days (and 7 days). These methods differ in their detection of fibrils, as ThT fluorescence amplitudes are often very sensitive to fibril morphology. Consistent with this idea, in both the tau fragment (295-330) (Fig. 4a) and tauRD (Fig. 4b) aggregation experiments, the ThT fluorescence amplitude is lower for S320F_I328S than S320F. The important conclusion here is that the behavior of S320F_I328S tracks in the same direction in vitro and in cells. The data suggests this, and the differences can be attributed to the design and strength of each assay, as the in vitro experiment is performed at a high concentration in isolation, while in cells, the aggregation process occurs in a very complex environment. A central question, however, that we are focusing on in current experiments is whether the fibril conformation of S320F (i.e., mechanism of aggregation) is the same in vitro and in cells. We are actively pursuing this question by determining cryo-EM structures of recombinant S320F fragments, tauRD, and FL and comparing them to conformations observed in cells and FTD-tau patients harboring the S320F mutation in tau (in collaboration with the van Swieten lab) (see figure below).

[REDACTED]

To complement our cryo-EM studies, we are performing extensive simulations to identify intermediates to understand the relationship of these interactions to the final state. In our mind, another important question is whether the gained nonpolar contacts are part of the fibril or do

they release the amyloid motif for aggregation. Our lab is also very interested in how cellular factors, including molecular chaperones, interact with tau-containing S320F. For example, in a previous study, we showed that DnaJC7 binds to aggregation-resistant tau conformations with higher affinity than a P301L mutant or an aggregation-prone monomer (Hou et al. Nat Comm 2021). We currently do not know how S320F changes chaperone binding to tau nor how the additional mutation I328S impacts chaperone recognition. This change, of course, is more complicated as chaperone binding to tau can be altered by mutations far from the binding site. Another possible consideration is how S320F or S320F_I328S alters binding to microtubules. Clearly, these are fascinating questions, and we are keen to understand the unique ability of the S320F among FTD-tau mutations to promote spontaneous aggregation in cells and how these efforts will gain insight into the regulation of tau misfolding into defined conformational states.

2. With the data in Fig 5E, what do the different "extents" of aggregation reflect based on max fluor values for ThT? Have the authors measured solubility by pelleting to better estimate the amount of aggregation?

We thank the reviewer for this comment. ThT can indeed yield different amplitudes as it binds more tightly to nonpolar and aromatic-rich surfaces on fibrils and thus changes in ThT affinity for fibril surfaces and stabilization of planar inter-ring geometries (PMID 19038267). Thus, it is not surprising that there is variation in ThT fluorescence amplitudes between different samples. Furthermore, given that the signal-to-noise is low in the experiment, we are confident that our samples have consistent ThT signal, albeit low. Given this type of signal variation across constructs, it is important to interpret the change in the context of kinetics; thus, our emphasis is to interpret these signal changes using $t_{1/2max}$ estimates to compare changes in the propensity to aggregate. Furthermore, supported by the presence of fibrils in the TEM images, we are confident that our samples form aggregates. We have adapted to improve visualization of the comparison of aggregation kinetics based on $t_{1/2max}$ values in Figures 5f and Supplementary Figs 5e and 6f. To complement our initial experiments on tauRD in vitro and in cells, we have carried out parallel experiments with tau peptides that span 295-330 (Supplementary Fig. 6e-g); consistent with our tauRD experiments, we observe variable ThT amplitudes in the curves. However, all yield fast kinetics as confirmed by $t_{1/2max}$ estimates

(Supplementary Fig. 6f). Finally, we confirm that we can detect fibrils in these samples using TEM even in conditions where the ThT signal is low (but has good signal-to-noise). Thus we are confident that samples that yield low ThT with good signal-to-noise are indeed able to aggregate spontaneously in vitro. These in vitro experiments fully agree with the spontaneous aggregation in cells, which is independent of the ThT signal. To directly address the reviewer's comment, we have performed a supernatant/pellet assay on tauRD S320F, tauRD S320I, tauRD S320I_I328V, and tauRD S320V. In this experiment, we aggregated the samples for 4 days and performed centrifugation at two speeds to separate tauRD into three species: 1) soluble monomer/oligomers, 2) small fibrils (100K spin) and 3) large fibrils (15K spin) (see figure below). As observed in the original experiments, S320F robustly forms fibrils in vitro, and the extent of the fraction of the sample forming fibrils for S320I_I328V is similar to S320F. The S320I and S320V tauRD constructs yielded a lower proportion of insoluble material, but consistent with our original TEM images, these samples still produce a positive signal in the pellet. In summary, the absolute ThT signal observed in an experiment is not directly proportional to the amount of aggregates present; thus, additional experiments (i.e., TEM) must be carried out to confirm the presence of aggregates. Combined with our new FL 2N4R experiments (Supplementary Figs. 5f,g and 6g,h), we are confident in the behavior of the S320F mutation, regulation of S320F aggregation via I328S and the additional computationally designed spontaneously aggregating S320I, S320V, S320I_I328V sequences across peptides, tauRD and FL 2N4R.

3. Related to the previous point, the fibrils appear a different morphology to the S320F mutant TEM (Fig. 5f and Supplementary Fig. 5e). Perhaps the fibrils undergo a very different assembly mechanism. Do they cross-seed each other's assembly?

We thank the reviewer for this comment. It is extremely difficult to interpret differences in the morphology of fibrils by TEM to infer changes in monomer conformation. For example, this is evidenced by SF and PHF conformations of tau fibrils in AD, which have identical monomer conformations but different morphology; thus, while the monomer fold may be identical, the way monomers interact may lead to different ultrastructures. To address this question, we are keenly attempting to determine fibril structures of S320F fibrils from in vitro preparations, cells, and tissues (see fig above). We have just made grids from one of the samples and are moving forward with data collection. Structural information is essential here to begin to understand how S320F induces tau to aggregation, whether the conformation from fragments and tauRD are the same, and how these conformations relate to cellular and patient aggregates. From our data, we hypothesize that the S320F induces rearrangement of protective nonpolar contacts that preferentially expose VQIVYK, but we do not yet know whether the gained nonpolar contacts will be observed in the final fibrillar assemblies. Structural information on these fibrils will be essential to begin to understand how to modify the aggregation mechanism. This study is just the beginning (of hopefully many) studies that will elucidate mechanisms of tau aggregation that will help us control tau fibril formation into distinct structures.

4. The percent of FRET positive cells for the 320V mutant is higher than for the 320F mutant - this enhanced aggregation differs to what was seen with purified protein (extent). How do the authors explain this? In my mind there are enough discrepancies between the data and model to make me question whether the model is justified.

Aggregation in vitro and in cells are distinct assays subject to vastly different conditions. Therefore, the range for the extent of aggregation is not equivalent; therefore, it is expected that these values are not identical. The promising results were the consistency of the overall trends, where sequences that do not aggregate in vitro do not aggregate in cells, and all samples that aggregated in vitro had enhanced aggregation in cells. As described in response to comment #1 by reviewer #2, it is very difficult to infer why there are differences in the aggregation in vitro and in cells because cells contain many proteins that interact with tau. We are unsure how these interactions are changed upon these mutations. In our calculations, we optimized the sequences for a specific fibril conformation without knowledge of other interactions, and thus this likely is the easiest explanation for these differences. As described above, complementary structural, computational studies validated by experiments will help explain some of these questions in subsequent studies.

Other comments:

1. It is not clear how the t50 values were extracted from the data. For example the data in Fig 1b has a complex shape. It is perhaps minor in the grand scheme of things, but it is hard to see how good fits to exponential or sigmoidal curves would be obtained. What is the explanation for the dip in ThT after about 12 hours?

We thank the reviewer for this comment. Plots were fitted to a non-linear regression model sigmoidal fit using Graphpad Prism, from which $t_{1/2max}$ values were derived. $t_{1/2}$ error represents a 95% CI. See figure for an example of the $t_{1/2max}$ fits of the aggregation curves for S320F₂₉₅₋₃₃₀ and S320I_I328V₂₉₅₋₃₃₀ tau peptides we reported in the manuscript to be 29.6 ± 1.6 and 51 ± 3.1 hrs.

Regarding ThT curves that reach a certain amplitude and then decrease over time, we fit the data to the highest amplitude. We suspect that the dip in ThT after about 12 hours might be explained by 1) dense molecular packing (i.e., bunding of fibrils that can change the overall affinity for ThT

by burying surfaces and 2) precipitation of large aggregates.

2. The S320F306-324 and WT306-324 fragments form amyloid almost instantly – do these also form fibrillar structures by electron microscopy? It remains possible that the mechanism of fibrillization is different when short peptides are used compared to the intact protein. One way to potentially test this would be to see if the fibrils of the peptides can seed the in-cell aggregation.

We appreciate the reviewer for suggesting a potential test. The peptides that span 306-324 behave very similarly to the VQIVYK amyloid motif alone because they are missing all the regulatory elements (DNIKHV on R2 and LGNIHH on R3). However, short peptides have been found not to seed very well in cellular models of tau aggregation because they form

supramolecular bundles, while for tauRD or FL tau, the termini help (i.e., fuzzy coat) limit secondary nucleation and elongation (PMID 31175300). This difference is indeed evident from TEM images of fragments compared to tauRD or FL in our experiments. Additionally, we have found that optimization of sonication can help but requires extensive times/powers. This requirement raises questions about whether fragmentation of the proteins is the dominant mechanism that leads to seeding. As part of determining cryo-EM structures, we have been optimizing the conditions to produce more singular structures. We will have a better handle on how to relate seeding to the supramolecular bundling of fragments.

Other issues:

1. The meaning of this sentence is unclear: “Most tauopathies are sporadic, linking the wildtype sequence of the microtubule-associated protein tau gene (MAPT) to each disease.” I presume the intended meaning is that wildtype tau forms deposits in the case of sporadic causes of disease.

We thank the reviewer for catching this, and we have clarified this sentence. Page 3, lines 6-7.

2. Repeated words in sentence: “and died aged 53 and died aged 53.”

We thank the reviewer for pointing this out. This typo has been fixed.

3. Something is missing in the following text as to why this suggests FTD-tau S325F is distinct to Pick’s disease “Immunohistochemistry of tau inclusions indicated presence of “Pick-like” bodies and “diffuse” staining. However, 3-repeat (3R) and 4-repeat (4R) tau were found in the insoluble tau fractions suggesting that the FTD-tau S320F tauopathy is distinct from that of Pick’s Disease.”

We thank the reviewer for the suggestion. We have modified this sentence in the manuscript to clarify that 3R and 4R tau were present in the inclusions, suggesting that the inclusions are distinct from Pick’s disease, which only contains 3R tau.

4. In this sentence, do the authors mean proximal to amyloid motifs or proximal to amyloids? If the latter, the sentence doesn’t make sense to me. If the former, some editing for clarity is needed. “This may suggest that S320F may allosterically influence amyloid motif-dependent aggregation, a mechanism distinct from other FTD-tau mutants proximal to amyloids.”

We thank the reviewer for pointing this out. We have corrected the sentence to clarify that we meant proximal to amyloid motifs.

5. Bottom of page 8 the authors cite Fig 4a – I think they mean to cite Fig 4b

We thank the reviewer for pointing this out. We have corrected the referencing to Fig. 4b.

6. On the top of page 9, the authors cite fibrils in Fig 4b. The electron microscopy (EM) data

is not shown in this panel. I presume they mean Fig S4E. I recommend that the EM data be provided in the main figure rather than supp.

We thank the reviewer for this suggestion. We have moved the TEM images of the tauRD aggregates into Fig 4.

Reviewer #4 (Remarks to the Author):

The article “FTD-tau S320F mutation stabilizes local structure and allosterically promotes amyloid motif-dependent aggregation” Chen et al describes an interdisciplinary research to elucidate the role of the pathological S320F mutation in enhancing the aggregation properties of Tau by triggering the exposure of the sequence 306-VQIVYK-311 in an allosteric manner.

The work presents an interesting mechanism that can potentially help understanding the underlying molecular bases of FTD.

Generally the research is conducted at high levels and spans a number of techniques in biophysics to study this pathological mutation.

I have, however, strong concerns on one of the central parts of the study, the MD simulations.

Major points:

1) Studying the properties of IDPs by MD simulations is a challenge, primarily for two reasons: the force field biases and the incompleteness of the samplings. In this study, the authors use AMBER99sb-ILDN. This force field is a decent one for folded proteins but it has some limitations in studying IDP, primarily because it is biased toward secondary structures and compact shapes. In general, MD simulations of IDPs can be heavily biased by the force field propensities (for example see 10.1021/acs.jctc.5b00736).

In order to make the MD part statistically significant, a second set of simulations of equivalent length should be run using an orthogonal force field. For example CHARMM22* with TIP4P-D waters is a good setup to reproduce NMR data of IDPs. A comparison between AMBER99sb and CHARMM22* simulations must therefore lead to the same conclusions in order to provide evidence for absence of force-field biases in the MD part.

We thank the reviewer for this suggestion. We have now carried out parallel simulations using the CHARMM27 force-field with the TIP4P-D model. These simulations were initiated from the same preminimized structure as in AMBER99sb-ILDN simulations, and a single replicate for each of the three peptides (WT₂₉₅₋₃₃₀, S320F₂₉₅₋₃₃₀, S320F I328S₂₉₅₋₃₃₀) was carried out for the equivalent 3us time. This simulation aimed to identify differences in the nonpolar clustering of

amino acids between WT and mutant, so we focused on the reconstitution of nonpolar residue clustering patterns as observed with the AMBER force-field. The interaction identified in AMBER was between the 320 and 328 positions as the S320F mutation introduces a new nonpolar residue that shifts interactions. We find that the interactions identified by the simulations using the AMBER99sb-ILDN force-field were recapitulated in the CHARMM27 replicate (Supplementary Fig. 4e-g). For the simulations, when comparing the interactions of the mutated S320 positions in the three peptide systems, we see an increased interaction with the I328 position in the S320F mutant relative to the WT peptide. Also, in the reversion mutant S320F I328S, we see a reduction in this interaction (Supplementary Fig. 4g). Additionally, we observed that the interaction distance between residues 320 and 328 followed distributions that were not statistically different from those seen in the 5 replicates of the AMBER99sb-ILDN simulations (Supplementary Fig. 4e-g). From the literature, we found that CHARMM22*, CHARMM27*, and CHARMM36* force-fields are biased toward helical secondary structure (PMID 35373198) despite starting with a known beta-hairpin conformation indicating that CHARMM* force-field may not be optimal to model beta strands and turns as well as AMBER*. Despite subtle bias in fluctuations of secondary structure, our comparison of AMBER99sb-ILDN and CHARMM27 simulations identify new nonpolar interactions driven by the S320F mutation. Our cumulative simulations highlight that independent of the details of the force-field, nonpolar residue clustering can be recapitulated, drawing similar conclusions. Importantly, the predictions from simulations using the AMBER99sb-ILDN and now in CHARMM27 force-field identify a region that, when mutated, reduces tau S320F-driven aggregation propensity in vitro and in cells.

2) The presented simulations aren't assessed for their convergence. If the simulations are not convergent, in addition to the force field dependencies, the results can be biased by the starting configurations as well as by the stochastically sampled non-converged conformations. Specific analyses are therefore required to check the convergence of the phase space exploration in the simulations. Moreover, the contact maps should also be calculated for the individual 5 simulations in each system (i.e. and not only as cumulative). This should prove convergence across the independent simulations.

We thank the reviewer for this suggestion. To address the convergence of the simulations performed using the AMBER force field, we computed 2D pairwise root mean square deviations (2D-RMSD) to assess simulation convergence towards equilibrium. This analysis additionally quantified the transition between structures over the simulation. We analyzed the wild type and mutants S320F and S320F_I328S trajectories, calculating 2D-RMSD plots for all five trajectories and reporting on two (of five)

representative trajectories (Supplementary Fig. 3b,f). In these plots, the color indicates the RMSD between conformation along time (ns) on the x-axis and y-axis. The RMSD values are calculated for each combination of structures within a trajectory and illustrate groups of conformations that share structural features highlighted by a low RMSD value (Supplementary Fig. 3b,f, blue color). The RMSD values ranged from 0 to 1.8 nm and were colored blue to red. Comparisons of the 2D-RMSD map of the wild-type and the two mutants suggest that S320F and S320F_I328S achieved the equilibrium after 1200ns and 1000 ns, respectively, with average RMSD of 0.8 -1.2 nm, while wild-type achieved equilibrium more quickly at around 500 ns with avg RMSD 0.8-1.0 nm. Visual inspection of the RMSD maps suggests that the mutation destabilized the structure initially and caused a greater extent of structural changes, which achieved a steady state later compared to wild-type. These analyses indicate that our simulations are converged. We also included the contact maps for each simulation (see full contact maps for all replicates below). These contact maps show similar interactions across the replicates, suggesting that the sampled states are seen across replicates. Briefly, we see interactions at the N-terminus of the peptide indicative of a B-hairpin-like conformation for WT and, to a lesser extent, for S320F and S320F_I328S. The S320F peptide also reveals more short-range interactions at the C-terminus that are shifted in the WT peptide and not present in the S320F_I328S reversion mutant. Although the contact maps are not identical across the replicates, the interaction similarities suggest that the initial structure is not the only bias for the sampling in the replicates.

3) The contact maps show some antiparallel pairings between consecutive stretches in the simulated Tau fragment (e.g. between DNIKHV and GGSVQI). These can be due to populations of beta-hairpin conformations in the ensemble, a known bias of force fields like AMBER99sb-ILDN. A plot of the secondary structure along the trajectories can elucidate this aspect (e.g. just by using `do_dssp` in GROMACS). If this plot shows some conformations featuring beta-hairpins, further analyses are required to validate the accuracy of the simulations. For example by comparing the results with experimental data in literature (e.g. secondary shifts in solution NMR of this fragment, if available).

We thank the reviewer for this suggestion. We have performed the DSSP secondary structure analysis for each of the replicates of WT, S320F, and S320F_I328S. Representative plots of two replicates are included in Supplementary Fig 3c,g. Overall, there is no clear bias towards a specific secondary structure in the AMBER simulations. Specifically, we see that the motif PGGG preferentially samples a beta-turn and the amyloid motif VQIVYK and, to a lesser extent, the DNIKHV region, sample extended, and beta-sheet conformations. These secondary structure preferences are to be expected and supported by experimental evidence from NMR experiments that suggest the DNIKHVPGGGVQIVYK element in tau samples specific conformations. The VQIVYK motif (commonly referred to as PHF*) contains 3 beta-branched residues and, in many experiments, has been shown to adopt a beta-sheet in solution (estimated to be ~25% from secondary shift analyses) (PMID 19226187, 15855160, 35373198). The PGGG motif is a known beta-turn stabilizing motif, and again experiments support that this sequence adopts conformations consistent with a turn (PMID 19226187, 15855160, 35373198). Finally, the DNIKHV sequence is less defined but again contains two beta-branched residues and thus has the capacity to adopt a beta-sheet conformation. This chemical shift information and SAXS and FRET data have also been used to build experimentally derived structural ensembles that recapitulate this transient structure (PMID 35373198). The structural preference for adopting these secondary structure conformations seen in our simulations is consistent with experimental data on the conformation of tau in solution. Interestingly, the CHARMM27 force-

field in our validation simulations showed helical bias in the VQIVYK amyloid motif sequence, which contradicts experiments. Indeed, CHARMM* has been shown to be biased towards helices (PMID 22904695). This highlights that simulations alone may be misleading and underscores the importance of experiments to validate the observations.

4) Another important test is to check for possible violations of the periodic boundary conditions. I understand that it is necessary to reduce as much as possible the box size for producing long runs, however, compact structures obtained after equilibration might easily expand during the simulations and therefore violate PBC.

We thank the reviewer for this suggestion. We performed the simulations in two stages: first, to obtain minimized initial structures from an extended random coil structure (5 replicates for WT and the 2 mutants, each 100ns) and second, for the final simulations to analyze the peptide behavior (every 3000 ns). For the second stage, we extracted the minimized structure from the first simulation and again prepared the system with a 1.2nm distance radius to the boundary which resulted in a reduced box size which improved performance for long production runs in the final simulation. We agree with the reviewer's comments. During the equilibration step, the peptide fragment compacts (from fully extended), and during the production run, the peptide fragment expands, increasing the chances of violating minimum image convention. The periodic boundary condition was applied, with the distance between the solute and the box of 1.2 nm in a dodecahedron box. GROMACS applied the periodic boundary conditions with minimum image convention, which implied that the cut-off radius terminated the non-bonded interactions that correspond to half the box length. Here, the cell was large enough, and the protein distance from the edge of the box was 1.2 nm which allowed the molecules to cross the boundaries at a sufficient distance from the next image so that no force calculations were made in between them (vdw cut-off =0.9 and coulomb cut-off =0.9). We used mean distance calculation to assess the distance from the minimum image convention. The minimum distance with periodic images (see figure) has been measured with avg and std dev for all replicates for each system and found the consistent shifts over the periodic boundary conditions.

Minor points

5) How were the termini of the peptides treated in the simulations?

The termini were not acetylated or amidated, but ions were used to neutralize the charge of the system.

6) Labels on figure 3 are partially missing.

We thank the reviewer for finding this error. We have corrected the panel labels in Figure 3.

REVIEWERS' COMMENTS

Reviewer #2 (Remarks to the Author):

The authors satisfactorily addressed all of my comments.

Reviewer #4 (Remarks to the Author):

The authors have addressed my previous points.

Convergence of the simulations as well as the agreement between CHARMM27 (TIP4P-D) and AMBER99sb-ILDN were not extremely high, however, considering the limitations of MD simulations of IDPs, this part of the research can be considered state of the art.

We thank the reviewers for their positive feedback on our revised manuscript and are excited that all reviewers agree that the manuscript is ready for publication in Nature Communications.

REVIEWERS' COMMENTS

Reviewer #2 (Remarks to the Author):

The authors satisfactorily addressed all of my comments.

We thank the reviewer for their positive assessment of our work.

Reviewer #4 (Remarks to the Author):

The authors have addressed my previous points.

Convergence of the simulations as well as the agreement between CHARMM27 (TIP4P-D) and AMBER99sb-ILDN were not extremely high, however, considering the limitations of MD simulations of IDPs, this part of the research can be considered state of the art.

We thank the reviewers for their positive assessment of our work.